# Guaranteed Discovery of Control-Endogenous Latent States with Multi-Step Inverse Models

## Abstract

In many sequential decision-making tasks, the agent is not able to model the full complexity of the world, which consists of multitudes of relevant and irrelevant information. For example, a person walking along a city street who tries to model all aspects of the world would quickly be overwhelmed by a multitude of shops, cars, and people moving in and out of view, each following their own complex and inscrutable dynamics. Is it possible to turn the agent's firehose of sensory information into a minimal latent state that is both necessary and sufficient for an agent to successfully act in the world? We formulate this question concretely, and propose the Agent Control-Endogenous State Discovery algorithm (`AC-State`), which has theoretical guarantees and is practically demonstrated to discover the *minimal control-endogenous latent state* which contains all of the information necessary for controlling the agent, while fully discarding all irrelevant information. This algorithm consists of a multi-step inverse model (predicting actions from distant observations) with an information bottleneck. `AC-State` enables localization, exploration, and navigation without reward or demonstrations. We demonstrate the discovery of the control-endogenous latent state in three domains: localizing a robot arm with distractions (e.g., changing lighting conditions and background), exploring a maze alongside other agents, and navigating in the Matterport house simulator.

## 1 Introduction

In many real-world systems, the observation space is generated by multiple complex and sparsely interacting subsystems. The state that is necessary for controlling an agent depends only on a small fraction of the information in the observation space. For example, consider a world consisting of a robot arm that is controlled by an agent with access to a high-resolution video of the robot. The agent's observation intertwines information that the agent controls with exogenous content such as lighting conditions or videos in the background. We define the *control-endogenous latent state* as the parsimonious representation, which includes only information that either can be controlled by the agent (such as an object on the table that can be manipulated by the arm) or affects the agent's control (e.g., an obstacle blocking the robot arm's motion). Discovering this representation while ignoring irrelevant information offers the promise of vastly improved planning, exploration, and interpretability.

A key challenge to discovering these representations in real world applications, is that only a continual stream of observations along with the agent's actions are available. For example, consider discovering the control-endogenous latent state from a video of a robot arm along with the sequence of actions taken to control the robot. Efroni et al. (2022c) demonstrated an algorithm for discovering control-endogenous latent state that uses open-loop planning with the agent reset to a fixed start state at the beginning of each episode, which is impractical in most real domains. Prior works (Efroni et al., 2022a; Wang et al., 2022b) also discovered the control-endogenous latent state by assuming access to a given factorized encoder, which is often not available in practice.

In this work, we consider a setting where the agent acts in a world consisting of complex observations with relevant and irrelevant information. We consider environments where observations are generated from states that can be decoupled into *control-endogenous* and *exogenous* components. Our definition of exogenous

strictly refers to aspects of the world that can never interact with the agent, which may include things like the detailed visual textures of objects or background processes that the agent cannot interact with. It is vital to discard irrelevant parts of the observation space. Our proposed algorithm predicts the first action that needs to be taken from any current observation to reach any future state in a certain number of steps in the future. We establish some theoretical results showing that this captures a complete and minimal control-endogenous state. This has many appealing properties in that we never need to learn a sequence model or a generative model. We only need to use supervised learning to predict actions. Prior works have considered partially observable settings to summarize agent-states (Littman et al., 1995; Kaelbling et al., 1998), or quantify agent states based on predictions of rewards (Littman et al., 2001). We emphasize here the salience of information for controlling the agent in fully-observed settings.

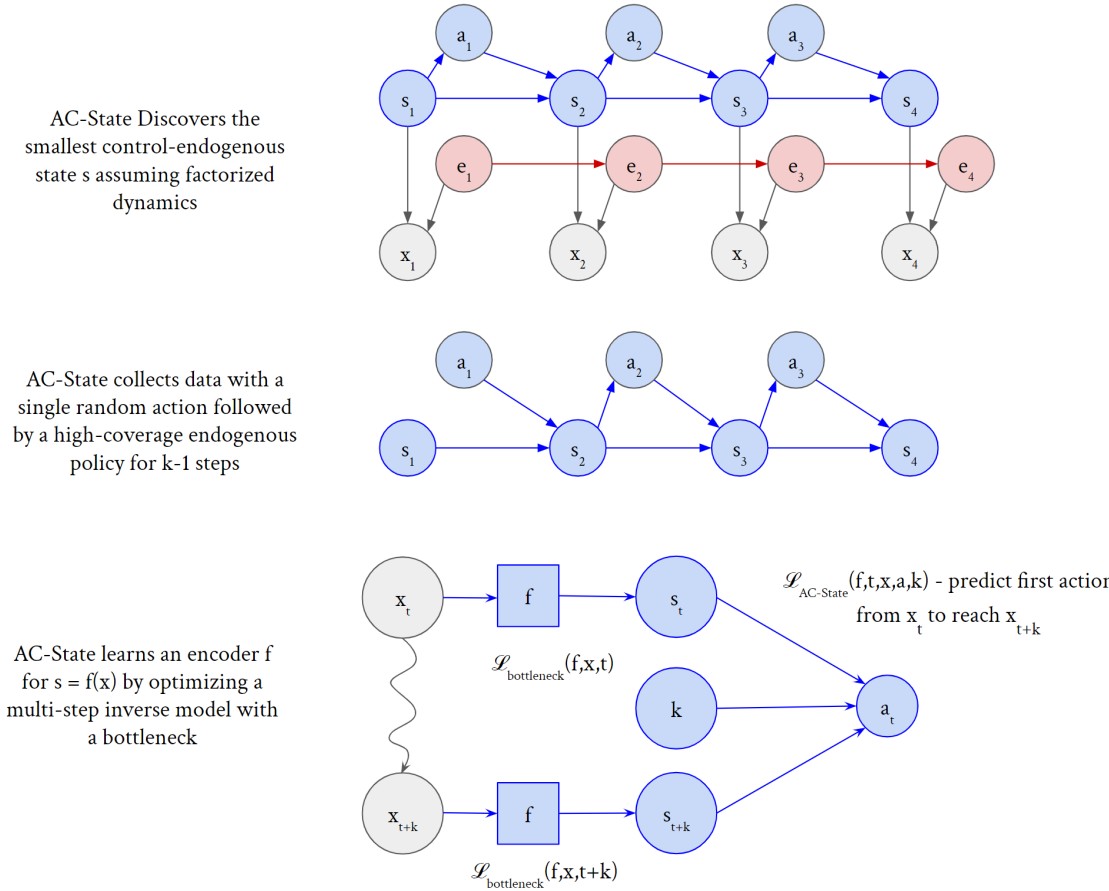

Figure 1: **Overview of `AC-State`**. The control-endogenous latent dynamics (top) can be discovered using `AC-State`, without learning a model of the exogenous noise $e$ or the observation $x$. This involves collecting data under a high-coverage endogenous policy (center) while optimizing a multi-step inverse model with a bottleneck (bottom).

How can we discover this control-endogenous latent state from raw observations and actions? We introduce the Agent Control-Endogenous (`AC-State`) algorithm, which provably guarantees discovery of the control-endogenous latent state by excluding all aspects of the observations that are unnecessary for control. `AC-State` learns an encoder $f$ that maps a given observation $x$ to the corresponding control-endogenous latent state $f(x)$. This is accomplished by optimizing a novel objective using an expressive model class such as deep neural networks (LeCun et al., 2015). The proposed algorithm uses a multi-step inverse model as its objective, along with a bottleneck on the capacity of the learned representation. As we discuss later, we use a discrete information bottleneck, based on vector quantization, which plays a key role in discarding exogenous information.

| Algorithms | PPE | OSSR | DBC | CDL | Denoised-MDP | 1-Step Inverse | AC-State (Ours) |
|---|---|---|---|---|---|---|---|
| Exogenous Invariant State | ✓ | ✓ | ✓ | ✓ | ✓ | ✓ | ✓ |
| Exogenous Invariant Learning | ✓ | ✓ | ✗ | ✗ | ✗ | ✓ | ✓ |
| Flexible Encoder | ✓ | ✗ | ✓ | ✗ | ✓ | ✓ | ✓ |
| YOLO (No Resets) Setting | ✗ | ✓ | ✓ | ✓ | ✓ | ✓ | ✓ |
| Reward Free | ✓ | ✓ | ✗ | ✓ | ✓ | ✓ | ✓ |
| Control-Endogenous Rep. | ✓ | ✓ | ✗ | ✓ | ✓ | ✗ | ✓ |

Table 1: **An Overview of the Properties** of prior works on representation learning in RL is shown, with a particular emphasis on robustness to exogenous information. We compare with several baselines including PPE Efroni et al. (2022c), OSSR Efroni et al. (2022a), DBC Zhang et al. (2021a) , Denoised MDP Wang et al. (2022a) and 1-Step Inverse Models Pathak et al. (2017). The comparison to `AC-State` aims to be as generous as possible to the baselines. ✗ is used to indicate a known counterexample for a given property.

In our experiments, we demonstrate the discovery of agent control-endogenous latent states that are (nearly) identical to their ground-truth values, while only using access to the raw observations and actions. We measure the exactness of the correspondence between the ground truth latent state (not used for training) and the learned latent state. The `AC-State` approach requires *no* reward signal, and many experiments require a few thousand interactions with the environment for initial learning along with *zero* additional examples to discover an appropriate policy. At the same time, successful discovery of the latent state implies the ability to localize, explore, and plan to reach goal states from a single shared representation.

## 2 Prior Approaches

Deep learning architectures can be optimized for a wide range of differentiable objective functions. Our key question is: what is an objective for provably learning a control-endogenous latent state that is compatible with deep learning? At issue is finding parsimonious representations that are sufficient for control of a dynamical system given observations from rich sensors (such as high-resolution videos) while discarding irrelevant details. Approaches such as optimal state estimation (Durrant-Whyte & Bailey, 2006), system identification (Ljung, 1998), and simultaneous localization and mapping (Cadena et al., 2016; Dissanayake et al., 2001) achieve parsimonious state estimation for control, yet require more domain expertise and design than is desirable. Previous learning-based approaches failed to capture the full control-endogenous latent state or exclude all irrelevant information (Efroni et al., 2022c). Reinforcement learning approaches that capture the latent state from rich sensors by employing autoencoders (Goodfellow et al., 2016) or contrastive learning often capture noise components[1].

Learning latent states for interactive environments is a mature research area with prolific contributions. We discuss a few of the most important lines of research and how they fail to achieve guaranteed discovery of the control-endogenous latent state. We categorize these contributions into broad areas based on what they predict: rewards (or value functions), future latent states, observations, the relationship between observations, and/or actions. Table 1 provides a summarized comparison of the properties of `AC-State` and prior works. In this table, algorithms that must learn the full exogenous state before learning how to discard it are marked as having an exogenous invariant state but not exogenous invariant learning.

**Limitations of Predicting Rewards:** Reward-based bisimulation (Zhang et al., 2021b) can filter irrelevant information from the latent state but is dependent on access to a reward signal. Deep reinforcement learning based on reward optimization (Mnih et al.) often struggles with sample complexity when the reward is sparse and fails completely when the reward is absent. An adversarial approach can also be used to learn disentangled reward-relevant and reward-irrelevant state (Fu et al., 2021). This can successfully ignore

---

[1]Consider a divided freeway, where cars travel on opposing sides of the lane. For this situation, autoencoders or contrastive learning objectives (commonly referred to as "self-supervised learning") produce distinct latent states for every unique configuration of cars on the other side of the tane divider. For example, an autoencoder or generative model would learn to predict the full configuration and visual details of all the cars, even those that could not interact with the agent.

distractors but may struggle if the distractors are difficult to completely model and can only be used when an external reward signal is available.

**Limitations of Predicting Latent States:** Approaches that involve predicting future latent states from past latent states have achieved good performance, but there is no theoretical guarantee that the latent state captures the full control-endogenous latent state (Guo et al., 2022; Schwarzer et al., 2021b; Ye et al., 2021; Pathak et al., 2017). Deep bisimulation approaches learn state representations for control tasks with agnosticism toward task-irrelevant details (Zhang et al., 2021a). Prior approaches have involved a model that predicts latent states such that the pre-trained representations can solve a task (Schwarzer et al., 2021a), yet this approach is not task-agnostic, and it is not guaranteed to recover the control-endogenous state. An auto-encoder trained with reconstruction loss or a dynamics model (Lange et al., 2012; Wahlström et al., 2015; Watter et al., 2015) learns low-dimensional state representations to capture information relevant for solving a task. Although learning latent state representations has been shown to be useful for solving tasks, there is no guarantee that such methods can fully recover the underlying control-endogenous latent state.

**Limitations of Predicting Observations:** Other works have also used generative models or autoencoders to predict future observations for learning latent representations, mostly for purposes of exploration. The idea of predicting observations is often referred to as "intrinsic motivation" to guide the agent towards exploring unseen regions of state space (Oudeyer & Kaplan, 2007). Other works use autoencoders to estimate future observations in the feature space for exploration (Stadie et al., 2015), though such models can also fail in the presence of exogenous observations. Dynamics models learn to predict distributions over future observations, but often for the purpose of planning a sequence of actions instead of recovering latent states or for purposes of exploration. Models which predict observations directly must also predict exogenous noise (Misra et al., 2020). Some approaches that learn generative models in the observation space attempt to learn a further decomposition of the dynamics into control-endogenous and exogenous latent states. While this can result in the correct latent state, it still requires learning a full generative model over the entire latent state, not just the control-endogenous latent state (Wang et al., 2022a;b).

**Limitations of Predicting Relationships between Observations:** By learning to predict relations between two consecutive observations, prior works have attempted to learn latent state representations, both theoretically (Misra et al., 2020) and empirically (Mazoure et al., 2020). For example, by exploiting mutual information based objectives (information gain based on current states and actions with future states) (Mazoure et al., 2020; Song et al., 2012), previous works have attempted to learn control-endogenous states in the presence of exogenous noise. However, unlike `AC-State`, they learn latent states dependent on exogenous noise, even though the learned representation can be useful for solving complex tasks (Mazoure et al., 2020). Theoretically, Misra et al. (2020) uses a contrastive loss based objective to provably learn latent state representations that can be useful for hard exploration tasks. However, contrastive loss based representations have a counterexample where they are forced to capture exogenous noise (Efroni et al., 2022c), whereas `AC-State` exploits an exogenous free rollout policy with a multi-step inverse dynamics model to provably and experimentally recover the full control-endogenous latent state.

**Limitations of Predicting Actions:** `AC-State` aims at recovering the control-endogenous latent states by training a multi-step inverse dynamics model in the presence of exogenous noise. While prior works have explored similar objectives, either for exploration or for learning state representations, they are unable to recover latent states with perfect accuracy in the presence of exogenous noise (Efroni et al., 2022c). In Appendix B, we provide a concrete counter-example demonstrating why a one-step inverse objective fails to learn the correct latent state. Approaches based on action prediction with one-step inverse models (Pathak et al., 2017) are widely used in practice (Baker et al., 2022; Badia et al., 2020). However, these inverse models can fail to capture the full control-endogenous latent dynamics (Efroni et al., 2022c; Hutter & Hansen, 2022), while combining them with an autoencoder (Bharadhwaj et al., 2022) inherits the weaknesses of that approach.

**Limitations of Empowerment-based objectives :** Empowerment-based objectives focus on the idea that an agent should try to seek out states where it is empowered by having the greatest number of possible states that it can easily reach (Klyubin et al., 2005). For example, in a maze with two rooms, the most

empowered state is the doorway, since it makes it easy to reach either of the rooms. Concrete instantiations of the empowerment objective may involve training models to predict the distribution of actions from observations (either single-step or multi-step inverse models) (Mohamed & Rezende, 2015; Yu et al., 2019), but they lack the information bottleneck term and the requirement of an exogenous-independent rollout policy. The analysis and theory in this work focus on action prediction as a particular method for measuring empowerment rather than as a way of guaranteeing the discovery of a minimal control-endogenous latent state and ignoring exogenous noise.

**Limitations of Causal Learning Approaches:** The concept of control-endogenous latent states is closely related to causality in the sense that it captures the content in the state space that is causally related to actions. This perspective has been explored by Wang et al. (2022b), which used conditional independence testing to find sets of high-level variables that are related to actions. Time series with known variables have also been used to discover causal relationships (Mastakouri et al., 2021). A pre-trained feature extractor for high-level variables (such as keypoints in computer vision) can also be used as the inputs for discovering causal structure (Li et al., 2020). These methods assume access to a small set of high-level variables with sparse causal structure. This assumption allows these methods to completely avoid the representational learning aspect of the problem. Learning causal dynamics end-to-end using deep learning is a nascent area of research, even when high-level variables are provided as the inputs for learning (Subramanian et al., 2022; Scherrer et al., 2021).

**Limitations of other approaches using Multi-Step Inverse Models:** Here we discuss several recent works that study the reinforcement learning problem in the presence of exogenous and irrelevant information. Efroni et al. (2022c) formulated the Ex-BMDP model and designed a provably efficient algorithm that learns the state representation. However, their algorithm succeeds only in the episodic setting, when the initial control-endogenous state is initialized deterministically. This is used alongside deterministic dynamics to use open-loop plans (i.e. plans which do not look at the actual state of the environment) to construct the endogenous latent state. These strict assumptions make their algorithm impractical in many cases of interest. Here we removed the deterministic assumption of the initial latent state. Indeed, as our theory suggests, `AC-State` can be applied in the "you-only-live-once" (YOLO) setting when an agent has access to a single trajectory. In Efroni et al. (2022b) and OSSR (Efroni et al., 2022a) the authors designed provable RL algorithms that efficiently learn in the presence of exogenous noise under different assumptions about the underlying dynamics. These works, however, focused on statistical aspects of the problem; how to scale these approaches to complex environments and combine function approximations is currently unknown and seems challenging. Efroni et al. (2022c) proposes a deterministic path planning based algorithm, which may not be directly compatible with most existing deep RL frameworks. In contrast, the proposed `AC-State` algorithm recovers the underlying latent states and can be directly used to construct a tabular MDP for planning, or for pre-training representations for later use in any deep RL based downstream task.

**Significance of our work:** `AC-State` uses a simple multi-step inverse model along with a bottleneck on the representations to learn latent states, which has not been exploited by prior works using dynamics models. Unlike the aforementioned works, here we focus on the representation learning problem. `AC-State` predicts actions based on future observations, and we show that a multi-step inverse dynamics model predicting actions can fully recover the control-endogenous latent states with no dependence on the exogenous noise. We emphasize that in the presence of exogenous noise, methods based on predictive future observations are prone to predicting both the control-endogenous and exogenous parts of the state space and do not have guarantees on recovering the latent structure. `AC-State` is the first practical and theoretically grounded approach that learns the control-endogenous representation with complex function approximators such as deep neural networks.

## 3 Exogenous Block MDP Setting

We consider the Exogenous Block Markov Decision Process (Ex-BMDP) setting to model systems with control-endogenous and exogenous subsystems, as formulated in Efroni et al. (2022c). This definition assumes that exogenous noise has no interaction with the control-endogenous state, which is stricter than the exogenous MDP definition introduced by Dietterich et al. (2018).

**Notation and Assumptions:** A BMDP consists of a set of observations, $\mathcal{X}$; a set of latent states, $\mathcal{Z}$; a finite set of actions, $\mathcal{A}$ with cardinality $A$; a transition function, $T : \mathcal{Z} \times \mathcal{A} \rightarrow \Delta(\mathcal{Z})$; an emission function $q : \mathcal{Z} \rightarrow \Delta(\mathcal{X})$; a reward function $R : \mathcal{X} \times \mathcal{A} \rightarrow [0,1]$; and a start state distribution $\mu_0 \in \Delta(\mathcal{Z})$. We do not consider the episodic setting, but only assume access to a single trajectory. The agent interacts with the environment, generating an observation and action sequence, $(z_1, x_1, a_1, z_2, x_2, a_2, \cdots)$ where $z_1 \sim \mu(\cdot)$. The latent dynamics follow $z_{t+1} \sim T(z' \mid z_t, a_t)$ and observations are generated from the latent state at the same time step: $x_t \sim q(\cdot \mid z_t)$. The agent does not observe the latent states $(z_1, z_2, \cdots)$, instead it receives only the observations $(x_1, x_2, \cdots)$. The *block assumption* holds if the support of the emission distributions of any two latent states are disjoint, $\mathrm{supp}(q(\cdot \mid z_1)) \cap \mathrm{supp}(q(\cdot | z_2)) = \emptyset$ when $z_1 \neq z_2$., where $\mathrm{supp}(q(\cdot \mid z)) = \{x \in \mathcal{X} \mid q(x \mid z) > 0\}$ for any latent state $z$. Lastly, the agent chooses actions using a policy $\pi : \mathcal{X} \rightarrow \Delta(\mathcal{A})$, so that $a_t \sim \pi(\cdot | x_t)$.

We now define the model we consider in this work, which we refer as *deterministic Ex-BMDP*:

**Definition 3.1** (Deterministic Ex-BMDP)**.** *A deterministic Ex-BMDP is a BMDP such that the latent state can be decoupled into two parts $z = (s, e)$ where $s \in \mathcal{S}$ is the control-endogenous state and $e \in \Xi$ is the exogenous state. We will further assume that the control-endogenous dynamics $T(s' \mid s, a)$ are deterministic. The exogenous dynamics $T_e(e' \mid e)$ may be stochastic. For $z, z' \in \mathcal{Z}, a \in \mathcal{A}$ the transition function is decoupled $T(z' \mid z, a) = T(s' \mid s, a) T_e(e' \mid e)$.*

The above definition implies that there exist mappings $f_\star : \mathcal{X} \rightarrow [S]$ and $f_{\star,e} : \mathcal{X} \rightarrow [E]$ from observations to the corresponding control-endogenous and exogenous latent states. The cardinality of the exogenous latent state $E$ may be arbitrarily large. Furthermore, we assume that the diameter of the control-endogenous part of the state space is bounded. In other words, there is an optimal policy to reach any state from any other state in a finite number of steps:

**Assumption 3.2** (Bounded Diameter of Control-Endogenous State Space)**.** *The length of the shortest path between any $z_1 \in \mathcal{S}$ to any $z_2 \in \mathcal{S}$ is bounded by $D$.*

We now describe a structural result of the Ex-BMDP model, proved in Efroni et al. (2022c). We say that $\pi$ is an *endogenous policy* if it is not a function of the exogenous noise. Formally, for any $x_1$ and $x_2$, if $f_\star(x_1) = f_\star(x_2)$ then $\pi(\cdot \mid x_1) = \pi(\cdot \mid f_\star(x_2))$.

Let $\mathbb{P}_\pi(s' \mid s, t)$ be the probability to observe the control-endogenous latent state $s = f_\star(x')$, $t$ time steps after observing $s' = f_\star(x)$ and following policy $\pi$, starting with action $a$. Let the exogenous state be defined similarly as $e = f_{\star,e}(x)$ and $e' = f_{\star,e}(x')$. The following result shows that, when executing an endogenous policy, the future $t$ time step distribution of the observation process conditioning on any $x$ has a decoupling property. Using this decoupling property we later prove that the control-endogenous state partition is sufficient to minimize the loss of the `AC-State` objective.

**Proposition 3.3** (Factorization Property under an Endogenous Policy, (Efroni et al., 2022c), Proposition 3)**.** *Assume that $x \sim \mu(x)$ where $\mu$ is some distribution over the observation space and that $\pi$ is an endogenous policy. Then, for any $t \geq 1$ it holds that:*

$$\mathbb{P}_\pi(x' \mid x, a, t) = q(x' \mid s', e') \mathbb{P}_\pi(s' \mid s, a, t) \mathbb{P}(e' \mid e, t).$$

Additionally, we assume that the initial distribution at time step $t = 0$ is decoupled $\mu_0(s, e) = \mu_0(e)\mu_0(e)$.

**Justification of Assumptions:** An intuitive justification for the concept of a control-endogenous latent state comes from the design of video games. A video game typically consists of a factorized game engine (which receives player input and updates a game state) and a graphics engine. Modern video game programs are often dozens of gigabytes, with nearly all of the space being used to store high-resolution textures, audio files, and other assets. A full generative model $T(x_{t+1}|x_t, a_t)$ would need to use most of its capacity to model these high-resolution assets. On the other hand, the core gameplay engine necessary for characterizing the control-endogenous latent dynamics $T(s_{t+1} \mid s_t, a_t)$ takes up only a small fraction of the overall game program's size and thus may be much easier to model[2]. To give a concrete example, when the 1993 game

---

[2]The control-endogenous latent state and their dynamics is generally even more compact than the game state, which itself may have exogenous noise such as unused or redundant variables.

*Link's Awakening* was remade in 2019 with similar gameplay but updated graphics, its file size increased from 8 MB to 5.8 GB, a 725x increase in size (Michel, 2022). Successful reinforcement learning projects for modern video games such as AlphaStar (Starcraft 2) and OpenAI Five (Dota 2) take advantage of this gap by learning on top of internal game states rather than raw visual observations (Vinyals et al., 2019; Berner et al., 2019). In the real world, there is no internal game state that we can extract. The control-endogenous latent state must be learned from experience.

Additionally, we make the further assumption that the data is collected under an endogenous policy. In this paper, we consider two kinds of policies: one is a random rollout policy, which is trivially an endogenous policy. The other type of policy we consider is achieved by planning to reach self-defined goals. This is achieved using a tabular-MDP constructed from counts of observed $(s, a, s')$ tuples, where $s$ is the discretized output of our learned encoder. The endogenous policy assumption in our theory may not always hold in practice, as a learned encoder may have errors that cause it to depend on exogenous noise.

We have also assumed that the control-endogenous state space has a finite state, a bounded diameter, and deterministic dynamics. The assumption of a finite state rules out continuous control problems and also makes problems with combinatorial structures (such as an arm with many joints or a robot interacting with multiple objects) infeasible. The requirement of a bounded diameter rules out environments that are unsafe or allow for irreversible changes, such as a robot becoming permanently stuck or damaged. The assumption of deterministic control-endogenous dynamics is not strictly required to use `AC-State`, but is required for a part of the proof that requires constructing the set of states that are reachable in a certain number of steps.

**Goal:** Our goal is reward-free and task-independent latent state discovery. Successful discovery of the control-endogenous latent state entails only learning a model for how to encode $s$ such that $T(s_{t+1} \mid s_t, a_t)$, while not learning anything about $T(e_{t+1} \mid e_t)$ or how to encode $e$. In particular, we seek to show that the control-endogenous latent state is identifiable and can be recovered exactly in the asymptotic regime.

## 4 AC-State: Discovering Agent Control-Endogenous Latent States

The control-endogenous latent state should preserve information about the interaction between states and actions while discarding irrelevant details. The proposed objective (Agent Control-Endogenous State or *AC-State*) accomplishes this by generalizing one-step inverse dynamics (Pathak et al., 2017; Efroni et al., 2022c; Hutter & Hansen, 2022; Badia et al., 2020) to multiple steps with an information bottleneck (Figure 1). This multi-step inverse model predicts the first action taken to reach a control-endogenous latent state $k$ steps in the future:

$$\mathcal{L}_{\texttt{AC-State}}(f, x, a, t, k) = -\log(\mathbb{P}(a_t | f(x_t), f(x_{t+k}); k)) \tag{1}$$

The multi-step inverse objective $\mathcal{L}_{\texttt{AC-State}}$ predicts the action $a_t$ from the observation just before the action $a_t$ is taken and from an observation collected $k$ steps in the future. The action step $t$ is sampled uniformly over all of the $\mathcal{T}$ steps in which the action was taken randomly. The prediction horizon $k$ is sampled uniformly from 1 to a maximum horizon $K_t$, which may be a fixed hyperparameter or may depend on the number of steps left in the planning horizon at step $t$. The multi-step inverse model predicts the distribution over the first action to reach a state $k$ steps in the future from a given current state. Generally, this action cannot be predicted perfectly, but it still carries information about the relationship between the current state and a state in the (potentially distant) future.

We optimize the parameters of an encoder model $f : \mathbb{R}^n \to \{1, \ldots, \text{Range}(f)\}$ that maps from an n-dimensional continuous observation to a finite latent state with an output range of size $\text{Range}(f)$, an integer number. $\mathcal{F}$ is a set that represents all mappings achievable by the model. We wish to minimize Objective 1 while using an encoder with the smallest latent state as its output. Conceptually, we can define this as finding a set of optimal solutions. The control-endogenous latent state is guaranteed to be one of the optimal solutions for the multi-step inverse objective $\mathcal{L}_{\texttt{AC-State}}$, while other solutions may fail to remove irrelevant information. The control-endogenous latent state is uniquely achievable by finding the lowest capacity $f$ in the set of optimal solutions to the `AC-State` objective:

$$G = \arg\min_{f \in \mathcal{F}} \mathbb{E}_{t \sim \mathcal{T}} \mathbb{E}_{k \sim U(1, K_t)} \mathcal{L}_{\texttt{AC-State}}(f, x, a, t, k) \tag{2}$$

$$\widehat{f} \in \arg\min_{f \in G} \mathrm{Range}(f). \tag{3}$$

As a concrete strategy to achieve a minimal latent state, we combine two mechanisms that are widely used in the deep learning literature to restrict the information capacity in $f$, which introduces an additional loss term $\mathcal{L}_{\mathrm{Bottleneck}}(f, x_t)$. We first pass the hidden state at the end of the network through a gaussian variational information bottleneck (Alemi et al., 2017), which reduces the mutual information between $x$ and the representation. We then apply vector quantization (Van Den Oord et al., 2017), which yields a discrete latent state. While either of these could be used on their own, we found that adding the gaussian mutual information objective eased the discovery of parsimonious discrete representations. Both of these are standard deep architectural elements, which we use along with their associated loss terms (Wang, 2020; Zeghidour et al., 2021). In-

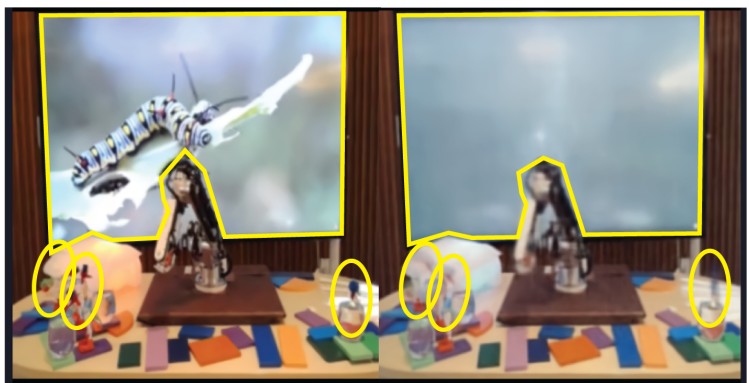

Figure 2: $\texttt{AC-State}$ **Successfully Removes Exogenous Noise:** In our experiments, we demonstrate an application of $\texttt{AC-State}$ in a Robot Learning Task. $\texttt{AC-State}$ discovers the control-endogenous latent state in a visual robotic setting with temporally correlated distractors: a TV, flashing lights, and drinking bird toys (left). The reconstruction from $\texttt{AC-State}$ decoder captures the arm's position, while ignoring all the irrelevant background distractors (right). This latent representation captures the relevant parts of the world without any supervision, reward signal, or hard-coded assumptions.

tuitively, the bottleneck is necessary to ensure that the latent contains the minimal amount of information necessary to obtain the optimal solution to the multi-step inverse model. In the absence of a bottleneck, we could set the encoder to be the identity function $\phi(x) = x$ and fail to learn a meaningful representation.

Since it avoids the need for complicated two-level optimization, we can consider using the bottleneck losses as additional terms in the objective. While we could re-weight these loss terms to make the bottleneck weaker, we did not find this to be necessary in practice. We optimize this approximate objective in practice:

$$\widehat{f} \approx \arg\min_{f \in \mathcal{F}} \mathbb{E}_{t \sim \mathcal{T}} \mathbb{E}_{k \sim U(1, K_t)} \left[ \mathcal{L}_{\texttt{AC-State}}(f, x, a, t, k) + \mathcal{L}_{\mathrm{Bottleneck}}(f, x_t) + \mathcal{L}_{\mathrm{Bottleneck}}(f, x_{t+k}) \right] \tag{4}$$

In addition to optimizing the objective and restricting capacity, the actions taken by the agent are also important for the success of $\texttt{AC-State}$, in that they must achieve high coverage of the control-endogenous state space and not depend on the exogenous noise. This is satisfied by a random policy or a policy that depends on $\widehat{f}(x_t)$ and achieves high coverage. The $\texttt{AC-State}$ objective enjoys provable asymptotic success (see section 5) in discovering the control-endogenous latent state.

Intuitively, the $\texttt{AC-State}$ objective encourages the latent state to keep information about the long-term effect of actions, which requires storing all information about how the actions affect the world. At the same time, the $\texttt{AC-State}$ objective never requires predicting information about the observations themselves, so it places no value on representing aspects of the world that are unrelated to the agent.

**AC-State with Random Policy:** The simpler version of AC-State uses a random policy to collect data and fit the encoder (Algorithm 1). While this is simple to implement, a uniform random policy is always exogenous-free by construction and asymptotically achieves high coverage. At the same time, it may not achieve sufficient coverage with reasonable sample efficiency for some hard exploration tasks. If a single bad action can cause an agent to lose many steps of progress, the sample complexity needed to achieve high coverage can scale exponentially in the horizon.

---

**Algorithm 1** AC-State Algorithm for Latent State Discovery Using A Uniform Random Policy

---

1: Initialize observation trajectory $x$ and action trajectory $a$. Initialize encoder $f_\theta$. Assume a control-endogenous diameter of $K$ and a number of samples to collect $T$, and a set of actions $\mathcal{A}$, and a number of training iterations $N$.
2: $x_1 \sim U(\mu(x))$
3: **for** t = 1, 2, ..., $T$ **do**
4: $\quad a_t \sim U(\mathcal{A})$
5: $\quad x_{t+1} \sim \mathbb{P}(x'|x_t, a_t)$
6: **for** n = 1, 2, ..., $N$ **do**
7: $\quad t \sim U(1, T)$ and $k \sim U(1, K)$
8: $\quad \mathcal{L} = \mathcal{L}_{\texttt{AC-State}}(f_\theta, t, x, a, k) + \mathcal{L}_{\text{Bottleneck}}(f_\theta, x_t) + \mathcal{L}_{\text{Bottleneck}}(f_\theta, x_{t+k})$
9: $\quad$ Update $\theta$ to minimize $\mathcal{L}$ by gradient descent.

---

**AC-State with Planning Policy:** A more involved algorithm (shown in Appendix C, Algorithm 2) is able to discover latent states in problems where exploration is difficult (such as in an environment where specific actions need to be taken in sequence to make progress). This can be accomplished by using a planning policy instead of a purely random policy. While any type of policy could be used in principle, we experimented with planning in a simple count-based tabular-MDP. In this model, a monte-carlo breadth-first search (suitable for stochastic dynamics) is used to select rarely seen reachable states as goal states. Then Dijkstra's algorithm is used to construct closed-loop plans to reach the desired goal state. All collected sample observations are collected into a buffer and their representations are recomputed under the current encoder network and the tabular-MDP is reconstructed on every 100th iteration. Only a single random action is taken on the first step of each goal-seeking trajectory, and it is this random action that is predicted with the `AC-State` objective. While the `AC-State` theory assumes deterministic dynamics, the tabular-MDP constructed using learned latent states often has stochastic dynamics due to errors in the learned encoder.

**Implementation Details:** We provide implementation details of the `AC-State` algorithm, with pseudo-code of the algorithm provided in C. The `AC-State` algorithm collects data with either a uniform random rollout policy or a Dijkstra's algorithm based planning policy, as discussed in most of our experiments. This is because a random policy ensures that the actions are completely exogenous free, irrespective of whether the observation consists of exogenous information. We then learn an encoder representation $\phi$ using the multi-step inverse dynamics objective as in equation 1. The encoded representation is then passed through a Vector Quantization (VQ-VAE) bottleneck such as to map the continuous representation into a set of discrete latent codes. The `AC-State` representation is learnt end to end using the overall objective, including the information bottleneck as in 4. We use the learnt discrete latents for visualizing the underlying structure of the MDP (Figure 6, right). The corresponding continuous representation is probed (using a separately learned decoder that predicts the ground truth state) to measure the accuracy of latent state recovery (figure 4). For each experiment, we include additional details of data collection and the evaluation metric to measure accuracy of latent discovery.

**Measuring Successful Discovery of Control-Endogenous Latent State:** The result of training with `AC-State` is a discrete graph where there is a node for every control-endogenous latent state with edges representing actions leading to a state-action node from which weighted outcome edges lead back to control-endogenous latent state nodes. We estimate the transition probabilities in this control-endogenous latent space by using counts to estimate $T(s_{t+1}|s_t, a_t)$. Once this control-endogenous latent state and associated transition distribution is estimated, we can directly measure how correct and how parsimonious these dynamics are whenever the ground-truth control-endogenous latent state is available (which is the case in all of our experiments). We measure an $L_1$-error on this dynamics distribution as well as the ratio of the number of learned latent states to the number of ground truth control-endogenous states.

## 5 Theoretical Analysis

We present an asymptotic analysis of `AC-State` showing it recovers $f_\star$, the control-endogenous state representation. The mathematical model we consider is the deterministic Ex-BMDP. There, the transition model

of the latent state decomposes into a control-endogenous latent state, which evolves deterministically, along with a noise term–the agent-irrelevant portion of the state. The noise term may be an arbitrary temporally correlated stochastic process. If the reward does not depend on this noise, any optimal policy may be expressed in terms of this control-endogenous latent state. In this sense, the recovered control-endogenous latent state is sufficient for achieving optimal behavior.

The proof involves two steps. In the first step, we show that a Bayes-optimal solution to the multi-step inverse model objective can be achieved without dependence on exogenous noise. In the second step, we show that the minimal representation minimizing the multi-step inverse model objective is the control-endogenous state.

## 5.1 High Level Intuition for Theory

Our theoretical justification first proceeds by showing that the optimal multi-step inverse model only depends on the control-endogenous state (Section 5.2) given our assumed factorization of the latent dynamics. Thus the control-endogenous latent state is sufficient for an optimal solution to the multi-step inverse model objective. In some sense, this establishes that the latent state does not need to be split any more than is necessary.

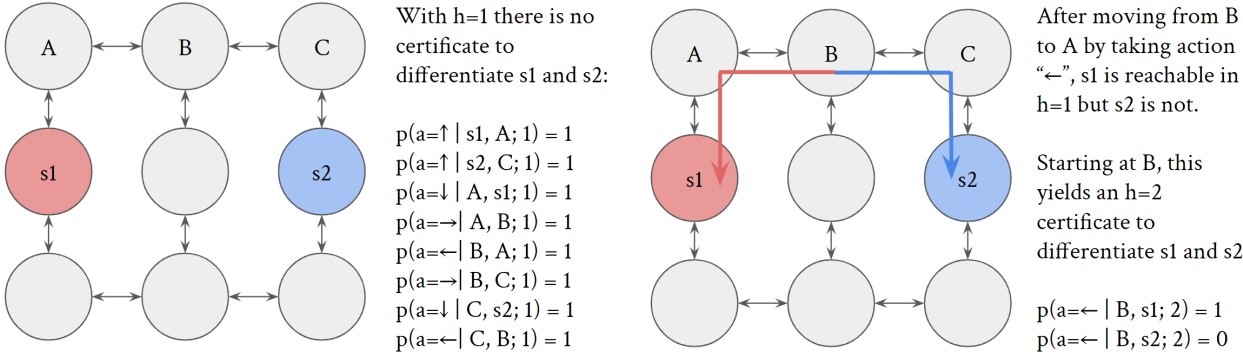

Figure 3: An illustration of the high-level idea of the proof in Section 5.3. The proof uses induction on $h$ to produce certificates which differentiate all pairs of distinct states. An example of one certificate using 2-step inverse models is shown on the right side of the figure. Going from $B$ to $s1$ requires the first action to move left (red arrow), whereas going to $s2$ requires that the first action moves right (blue arrow).

The next part of the argument (Section 5.3 and Figure 3) attacks from the opposite direction by showing that the multi-step inverse model forces the latent state to split any two states which are truly distinct. This means that there is no coarser solution to the multi-step inverse model objective (since the bottleneck will prefer the coarsest solution). The proof uses induction on the horizon length to produce certificates to split increasingly distantly-separated states. This novel result is somewhat counter-intuitive in a few ways. One surprising result is that it only requires the model to predict the first action taken to reach a future state, rather than predicting the full sequence of actions. Another interesting result is that the policy following the first action can be fairly general (needs to achieve high state coverage over the dataset) and includes a purely random rollout policy. The optimal solution to the multi-step inverse model objective will change depending on the choice of rollout policy, but the theory indicates that a wide variety of choices will lead to learning the correct representation.

## 5.2 Bayes Optimal Solution does not depend on Exogenous Noise

Consider the generative process in which $x$ is sampled from a distribution $\mu$, the agent executes a policy $\pi$ for $t$ time steps and samples $x'$. Denote by $\mathbb{P}_{\pi,\mu}(x, x', t)$ as the joint probability, and by $\mathbb{P}_{\pi,\mu}(a \mid x, x', t)$ as the probability that under this generative process the action upon observing $x$ is $a$. The following result, which builds on Proposition 3.3, shows that the optimal bayes solution $\mathbb{P}_{\pi,\mu}(a \mid x, x', t)$ is equal to $\mathbb{P}_{\pi,\mu}(a \mid f_\star(x), f_\star(x'), t)$ for $\mathbb{P}_{\pi,\mu}(x, x', t) > 0$, where $\mathbb{P}_{\pi,\mu}(x, x', t) > 0$ is the probability to sample $x$ .

**Proposition 5.1.** *Assume that $\pi$ is an endogenous policy. Let $x \sim \mu$ for some distribution $\mu$. Then, the Bayes' optimal predictor of the action-prediction model is piece-wise constant with respect to the control-endogenous partition: for all $a \in \mathcal{A}$, $t > 0$ and $x, x' \in \mathcal{X}$ such that $\mathbb{P}_{\pi, \mu}(x, x', t) > 0$ it holds that*

$$\mathbb{P}_{\pi, \mu}(a \mid x, x', t) = \mathbb{P}_{\pi, \mu}(a \mid f_\star(x), f_\star(x'), t).$$

We comment that the condition $\mathbb{P}_{\pi, \mu}(x, x', t) > 0$ is necessary since otherwise the conditional probability $\mathbb{P}_{\pi, \mu}(a \mid x, x', t)$ is not well defined.

The proof for this proposition is a straightforward application of bayes theorem, is given in the appendix, and was previously presented in Efroni et al. (2022c;a).

## 5.3 Discovery of Full Control-Endogenous Latent State

Proposition 5.1 from the previous section shows that the multi-step action-prediction model is piecewise constant with respect to the partition induced by the control-endogenous states $f_\star : \mathcal{X} \to [S]$. In this section, we assume that the executed policy is an endogenous policy that induces sufficient exploration. With this, we prove that there is no coarser partition of the observation space such that the set of inverse models is piecewise constant with respect to it.

We assume that the Markov chain induced on the control-endogenous state space by executing the policy $\pi_{\mathcal{D}}$ by which *AC-State* collects the data has a stationary distribution $\mu_{\mathcal{D}}$ such that $\mu_{\mathcal{D}}(s, a) > 0$ and $\pi_{\mathcal{D}}(a \mid s) \geq \pi_{\min}$ for all $s \in \mathcal{S}$ and $a \in \mathcal{A}$. We consider the stochastic process in which an observation is sampled from a distribution $\mu$ such that $\mu(s) = \mu_{\mathcal{D}}(s)$ for all $s$. Then, the agent executes the policy $\pi_{\mathcal{D}}$ for $t$ time steps. For brevity, we denote the probability measure induced by this process as $\mathbb{P}_{\mathcal{D}}$. A more detailed justification for this assumption is given in the appendix.

We begin by defining several useful notions. We denote the set of reachable control-endogenous states from $s$ in $h$ time steps as $\mathcal{R}_h(s)$.

**Definition 5.2.** (Reachable Control-endogenous States). *Let the set of reachable control-endogenous states from $s \in \mathcal{S}$ in $h > 0$ time steps be $\mathcal{R}(s, h) = \{s' \mid \max_\pi \mathbb{P}_\pi(s' \mid s, h) = 1\}$.*

Observe that every reachable state from $s$ in $h$ time steps satisfies that $\max_\pi \mathbb{P}_{\pi, \mu}(s' \mid s_0 = s, h) = 1$ due to the deterministic assumption of the control-endogenous dynamics.

Next, we define a notion of *consistent partition* with respect to a set of function values. Intuitively, a partition of space $\mathcal{X}$ is consistent with a set of function values if the function is piece-wise constant on that partition.

**Definition 5.3.** (Consistent Partition with respect to $\mathcal{G}$). *Consider a set $\mathcal{G} = \{g(a, y, y')\}_{y, y' \in \mathcal{Y}, a \in \mathcal{A}}$ where $g : \mathcal{A} \times \mathcal{Y} \times \mathcal{Y} \to [0, 1]$. We say that $f : \mathcal{Y} \to [N]$ is a* consistent partition *with respect to $\mathcal{G}$ if for all $y, y'_1, y'_2 \in \mathcal{Y}$, $f(y'_1) = f(y'_2)$ implies that $g(a, y, y'_1) = g(a, y, y'_2)$ for all $a \in \mathcal{A}$.*

Observe that Proposition 5.1 shows that the partition of $\mathcal{X}$ according to $f_\star$ is consistent with respect to $\{\mathbb{P}_{\mathcal{D}}(a \mid x, x', h) \mid x, x' \in \mathcal{X}, h \in [H] \text{ s.t. } \mathbb{P}_{\mathcal{D}}(x, x', h) > 0\}$, since, by Proposition 5.1, $\mathbb{P}_{\mathcal{D}}(a \mid x, x', h) = \mathbb{P}_{\mathcal{D}}(a \mid f_\star(x), f_\star(x'), h)$.

Towards establishing that the coarsest abstraction according to the `AC-State` objective is $f_\star$ we make the following definition.

**Definition 5.4.** (The Generalized Inverse Dynamics Set AC$(s, h)$). *Let $s \in \mathcal{S}, h \in \mathbb{N}$. We denote by $\mathrm{AC}(s, h)$ as the set of multi-step inverse models accessible from $s$ in $h$ time steps. Formally,*

$$\mathrm{AC}(s, h) = \{\mathbb{P}_{\mathcal{D}}(a \mid s', s'', h') : s' \in \mathcal{R}(s, h - h'), s'' \in \mathcal{R}(s', h'), a \in \mathcal{A}, h' \in [h]\}. \tag{5}$$

Observe that in equation equation 5 the inverse function $\mathbb{P}_{\mathcal{D}}(a \mid s', s'', h')$ is always well defined since $\mathbb{P}_{\mathcal{D}}(s', s'', h') > 0$. It holds that $\mathbb{P}_{\mathcal{D}}(s', s'', h') = \mathbb{P}_{\mathcal{D}}(s')\mathbb{P}_{\mathcal{D}}(s'' \mid s', h') > 0$, since $\mathbb{P}_{\mathcal{D}}(s') > 0$ and $\mathbb{P}_{\mathcal{D}}(s'' \mid s', h') > 0$. The inequality $\mathbb{P}_{\mathcal{D}}(s') > 0$ holds by the assumption that the stationary distribution when following

$\mathcal{U}$ has positive support on all control-endogenous states (Appendix D.3). The inequality $\mathbb{P}_{\mathcal{D}}(s'' \mid s', h') > 0$ holds since, by definition $s'' \in \mathcal{R}(s, h')$ is reachable from $s'$ in $h'$ time steps; hence, $\mathbb{P}_{\mathcal{D}}(s'' \mid s', h') > 0$ by the fact that the stationary assumption (Appendix D.3) implies that the policy $\pi_{\mathcal{D}}$ induces sufficient exploration.

**Theorem 5.5.** ($f_\star$ *is the coarsest partition consistent with respect to* AC-State *objective*). *Given the bounded diameter Assumption 3.2 and the high-coverage policy assumption (Appendix D.3) hold, it follows that there is no coarser partition than* $f_\star$, *which is consistent with* $\mathrm{AC}(s, D)$ *for any* $s \in \mathcal{S}$.

The proof of this theorem, which uses induction on $h$, is provided in the appendix.

## 6 Experiments

**Summary of Experimental Results:** Our experiments explore three domains and demonstrate the unique capabilities of `AC-State`: (i) `AC-State` learns the control-endogenous latent state of a real robot from a high-resolution video of the robot with rich temporal background structure (a TV playing a video, flashing lights, dipping birds, and even people) *occluding the information in the observations*; (ii) `AC-State` learns about the controlled agent while ignoring other random agents in a multiple maze environment where there are other functionally identical agents. We demonstrate that `AC-State` solves a hard maze exploration problem. (iii) `AC-State` learns a control-endogenous latent state in a house navigation environment where the observations are high-resolution images and the camera's vertical position randomly oscillates, showing that `AC-State` is invariant to exogenous viewpoint noise that radically changes the observation.

### 6.1 Robotic Arm Experiment

We first evaluate `AC-State` to recover the latent states in a real robot arm experiment consisting of observations with exogenous information in the background. We briefly describe the experiment setup and the evaluation metric for measuring latent state recovery. We found that `AC-State` discovers the control-endogenous latent state of a real robot arm while ignoring distractions.

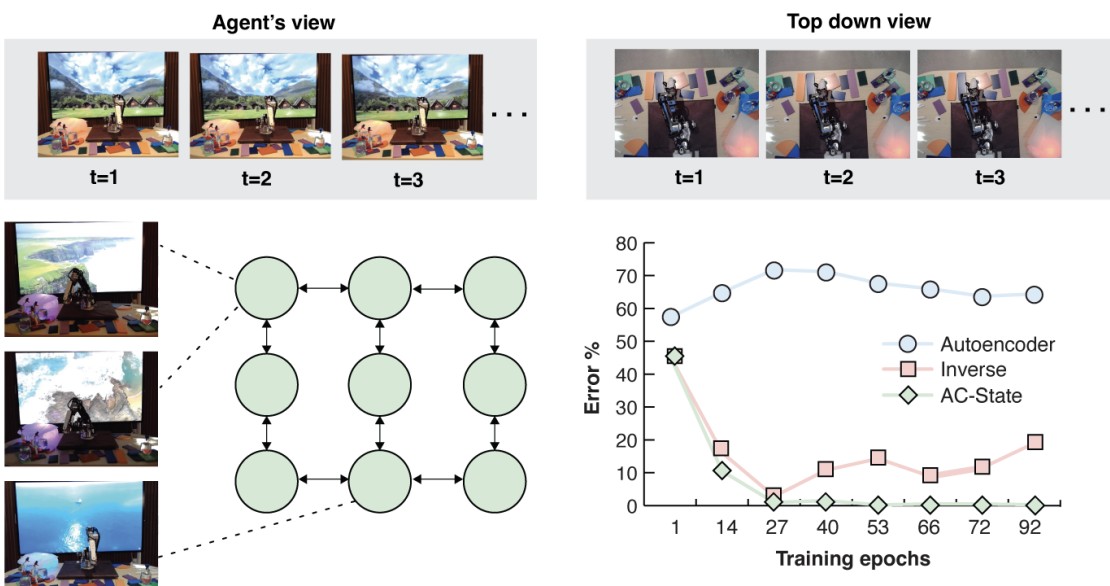

Figure 4: A real robotic arm moves between nine different positions (left). The four actions correspond to the arm moving forward, backward, to the left, or to the right. The quality of control-endogenous latent state dynamics learned by `AC-State` is better than one-step inverse models and autoencoders, as measured by the dynamics difference error (bottom right).

**Experiment Setup:** We collected 6 hours of data (14,000 samples) from an AR-3 robot arm (Annin, 2022) by taking give high-level actions (move left, move right, move up, move down and stay). A picture of the robot was taken after each completed action. We used a maximum horizon of $K = 5$ corresponding to the diameter of the endogenous part of MDP, but empirically observed that larger horizons also worked

well. The observations were mapped to a representation by using a 6-layer vision transformer (Dosovitskiy et al., 2021) with 4-heads per-layer. The action prediction for `AC-State` used a simple MLP on top of these representations and an embedding for $k$. Details of our experiment setup are provided in Appendix A.3. The robot was placed among many distractions, such as a television, flashing, color-changing lights, and people moving in the background. Some of these distractions (especially the TV) had strong temporal correlations between adjacent time steps, as is often the case in real-life situations.

**Experiment Result and Evaluation :** Figure 4 shows that `AC-State` can discover the control-endogenous latent state which is the ground truth position of the robot (not used during training). Qualitatively, we measure the accuracy of latent recovery of `AC-State` by training a convolutional neural network to reconstruct $x$ from $f(x)$ with square loss as the error metric. In other words, we learn a visualization of the latent state to map from the latent state $f(x_t)$ to an estimate $\hat{x}_t$ the observation $x_t$ by optimizing the mean-square error reconstruction loss $||\hat{x} - x_t||^2$. We found that the robot arm's position was correctly reconstructed, while the distracting TV and color-changing lights appeared completely blank, as expected (Figure 2). As discussed previously, an auto-encoder trained end-to-end with $x$ as input captures both the control-endogenous latent state and distractor noise. The theoretical counter-examples for autoencoders and one-step inverse models also closely match the errors that we observe experimentally (Figure 5). Our results show that `AC-State` can perfectly recover control-endogenous latent states, both qualitatively and quantitatively, in a practical robot arm task consisting of exogenous information in the background.

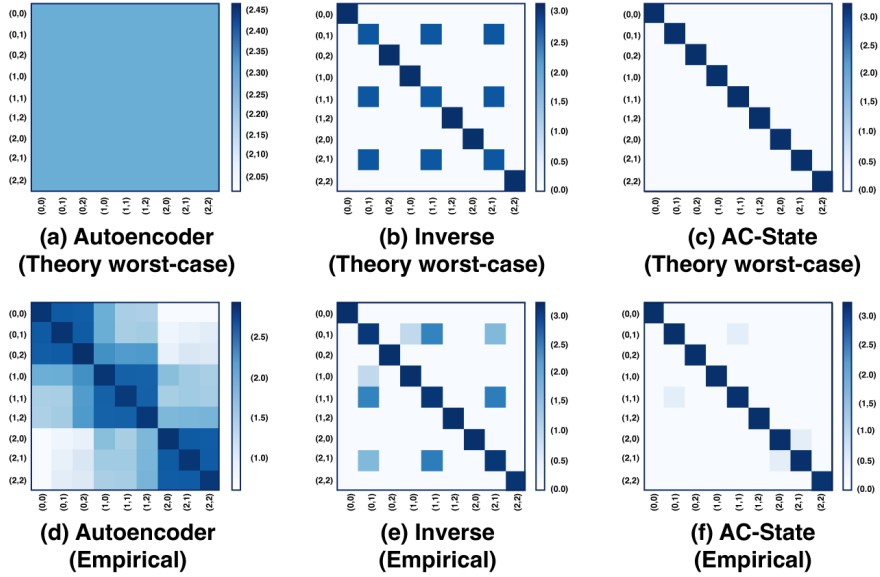

Figure 5: In co-occurrence histograms measuring performance in the robotic arm environment, autoencoders fail (left), one-step inverse models fall prey to the same counterexample in theory and in experiment (center), and `AC-State` discovers a perfect control-endogenous latent state (diagonal histogram, right). Intriguingly, the failure case of the one-step inverse model is the merging of three distant states, which are the arm being on the center-left, center, or center-right positions. These states being merged by a one-step inverse model is precisely what the theory predicts. On the other hand, the autoencoder merges states where the robot arm is in similar positions because the observations are visually similar.

## 6.2 Mazes with Exogenous Agents and Reset Actions

In our experiments, `AC-State` removes exogenous noise in environments where that noise has a rich and complex structure. We studied a multi-agent system in which a single agent is control-endogenous and the other agents follow their own independent policies. To do this, we take a pixel-based visual gridworld consisting of multiple mazes and agents, where the goal of `AC-State` is to recover only the controllable part of the maze while ignoring observations from other agents. In our experiments, we again qualitatively

show that `AC-State` can perfectly recover the underlying structure of the endogenous, controllable maze. Additional quantitative results measuring accuracy of latent state recovery is provided in appendix A.2.

**Experiment Setup:** In an environment with 9 agents and each agent having $c$ control-endogenous states, the overall size of the observation space is $c^9$. With 3,000 training samples, `AC-State` is able to nearly perfectly discover the agent's control-endogenous latent state, while fully ignoring the state of the 8 uncontrollable exogenous agents, with all of the agents controlled by a random policy (Figure 6). When using a random policy, the maximum horizon $K$ is set to the diameter of the MDP, whereas when using `AC-State`, the maximum horizon is based on the number of steps to the goal estimated by the planning process. Additional details on the multi-maze environment is provided in appendix A.2.

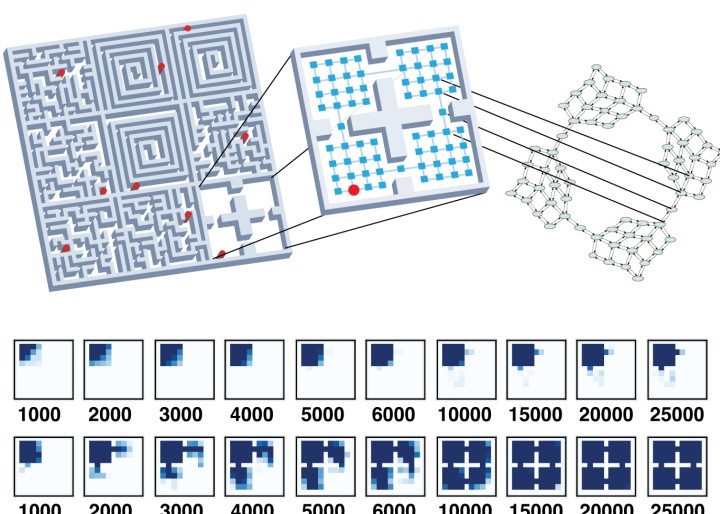

Figure 6: We study a multi-agent world where each of the nine mazes has a separately controlled agent. Training `AC-State` with the actions of a specific agent discovers its control-endogenous latent state while discarding information about the other mazes (top). In a version of this environment where a fixed third of the actions cause the agent's position to reset to the top-left corner, a random policy fails to explore (top heatmap sequence), whereas planned exploration using `AC-State` reaches all parts of the maze (bottom heatmap sequence).

**Experiment Result and Evaluation:** In the multi-maze system consisting of exogenous observation from other agents, we first evaluate the ability of `AC-State` to recover the underlying structure of the controllable states. Figure 6 shows that using the `AC-State` objective followed by the discretization bottleneck, we can visualize the endogenous states with a set of discrete latent codes using the quantization. Figure 6 (top-left) shows the observation of the agent with the controllable maze (top-middle) highlighted. Figure 6 (top-right) shows the extracted visualization with the discrete codes, qualitatively showing that endogenous-states can be recovered to retrieve the structure of the controllable MDP. Our goal is to show that the proposed `AC-State` agent can "discover" the control-endogenous latent state within the global gridworld while ignoring the structural perturbations in the geometry of the other 8 gridworlds. A comparison with baselines is shown in Table 2.

| Algorithms | True Endogenous States Explored | Dynamics Difference Error (%) |
|---|---|---|
| Contrastive | 20/68 | 92.2 |
| Autoencoder | 20/68 | 91.7 |
| Forward Generative | 25/68 | 97.1 |
| AC-State | **68/68** | **0.00** |

Table 2: Multiple-Maze Experiment Result with Baseline Comparisons, with each model having 25000 samples to explore in the environment. The baseline methods fail to ignore exogenous noise, fail to learn an effective tabular representation, and thus fail to explore effectively.

We then show that the control-endogenous latent state is useful when it allows for exponentially more efficient exploration than is achievable under a random policy. To exhibit this, we modified the maze problem by giving the agent additional actions that reset to a fixed initial starting position. When a third of all actions cause resets, the probability of acting randomly for $N$ steps without resetting is $(2/3)^N$. We show that a learned exploration policy using `AC-State` succeeds in full exploration and learning of the control-endogenous latent state with 25,000 samples, while a random exploration policy barely explores the first room with the same number of samples (Figure 6).

The counts of the discrete latent states are used to construct a simple tabular MDP where planning is done to reach goal states using a monte carlo version of Dijkstra's algorithm (to account for stochastic transition dynamics). The reachable goal states are sampled proportionally to $\frac{1}{count(s_i)}$, so the rarely seen states are the most likely to be selected as goals. Experiment results demonstrate that a goal-seeking policy achieves perfect coverage of the state space by using discovered latents for exploration, while a random policy fails to reach more than 25% of the state space in the presence of reset actions. We demonstrate this with heatmaps showing state visitation frequencies.

### 6.3 First-Person Perspective House Navigation

Finally, we evaluate `AC-State` in a simulated household navigation task where data is pre-collected from different positions in the household. We compare `AC-State` with several baselines and evaluate the ability to capture the endogenous latent state, ie, position of the agent in the house while discarding other irrelevant information captured by the high resolution camera. Additional details on the matterport simulator, along with experiment setup and implementation details, are provided in appendix A.4.

**Experiment Setup :** In order to analyze the performance of the proposed `AC-State` objective in a more realistic setting, we evaluated it on Matterport (Chang et al., 2017), a navigation environment where each observation is a high resolution image taken from a position in a real house. A 20,000 sample dataset collected from an agent moving randomly through the house is used to train `AC-State`. In addition to the high degree of visual information in the input observations, we randomly move the camera up or down at each step as a controlled source of irrelevant information (exogenous noise). `AC-State` removes the view noise from the encoded representation $f(x)$ while still capturing the true control-endogenous latent state (Figure 7), whereas other baselines capture both the control-endogenous latent state and the exogenous noise. The first-person house images are mapped to representations using a 6-layer, 4-head vision transformer (Dosovitskiy et al., 2021). The model is trained end-to-end with the `AC-State` objective with maximum

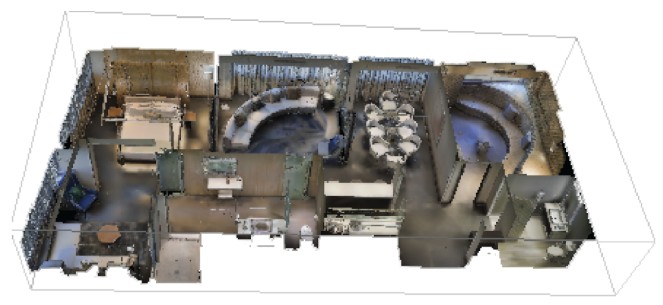

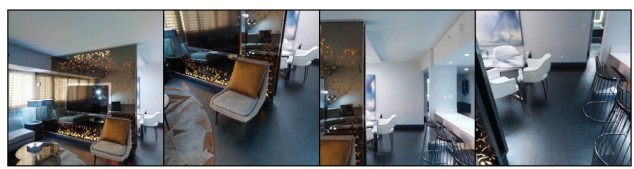

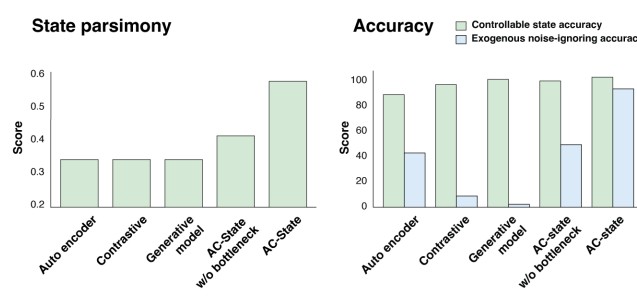

Figure 7: We evaluate `AC-State` in a house navigation environment (top), where the agent observes high-resolution images of first-person views and the vertical position of the camera is exogenous (center). `AC-State` discovers a control-endogenous latent state that is parsimonious (bottom left) and captures the position of the agent in the house while discarding the exogenous noise (bottom right). The baselines capture the control-endogenous latent state but fail to discard the exogenous noise.

horizon of $K = 5$ for 20 epochs using the Adam optimizer with a learning rate of 1e-4. We ran three seeds and report mean and standard deviation. Details on the matterport simulator are provided in Appendix A.4.

**Experiment Results and Evaluation :** We present the results for this experiment in Figure 7. The *Controllable Latent State Accuracy* is the viewpoint prediction accuracy for the current state. The *Exogenous Noise-Ignoring Accuracy* reflects how much information about the exogenous noise is removed from the observation. We can see that the proposed *AC-State* model has the highest control-endogenous latent state and exogenous noise-ignoring accuracy. Thus, it outperforms the baselines we considered at capturing control-endogenous state while ignoring exogenous noise. We calculated state parsimony as $\frac{\text{Num. Ground Truth States}}{\text{Num. Discovered States}}$. Therefore, a lower state parsimony denotes a high number of discovered states, which means that the model

fails at ignoring exogenous information. The proposed model has the highest state parsimony, which shows the effectiveness of the model in ignoring exogenous noise while only capturing control-endogenous latent states.

## 7 Conclusion

`AC-State` reliably discovers the control-endogenous latent state across multiple domains. The vast simplification of the control-endogenous latent state discovered by `AC-State` enables visualization, exact planning, and fast exploration. The field of self-supervised reinforcement learning particularly benefits from these approaches, with `AC-State` useful across a wide range of applications involving interactive agents as a self contained module to improve sample efficiency given any task specification. As the richness of sensors and the ubiquity of computing technologies (such as virtual reality, the internet of things, and self-driving cars) continues to grow, the capacity to discover agent-control endogenous latent states enables new classes of applications.

**Limitations and Future Work:** In this paper, we showed that `AC-State` can recover the control endogenous latent states with perfect accuracy, and recover the underlying latent tabular MDP by completely discarding the exogenous information. However, one limitation is that the success of `AC-State` depends on the use of the discrete information bottleneck and only applicable for environments with deterministic endogenous dynamics. An interesting extension of `AC-State` would be to derive an algorithm, with provable guarantees for stochastic environments and starting states. Furthermore, it would be interesting to see how the learnt representations from `AC-State` can be useful for the downstream RL task in presence of exogenous information. Since `AC-State` recovers the full underlying model, it can potentially be used for reaching any goal state in the exploration frontier, making the algorithm highly applicable for practical tasks such as household navigation.

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

# A   Methods and Experiments

## A.1   Dynamics Difference Error Metric

For every pair of learned state and action $(s, a)$, we have an empirical count-based distribution over next learned states $\mathbb{P}(s'|s, a)$. We can also empirically estimate the distribution over ground states for each learned state, which we can call $\mathbb{P}(g|s)$. We can sample over this to get an expected dynamics over the ground states translated from the learned state dynamics. We can also look at the empirical transition distribution over ground states. We then compute an L1-difference between these two distributions and uniformly average over all pairs of ground state and action values and refer to this as the dynamics difference error.

In this section, we will describe our methods and experiments for validating the proposed latent state discovery with `AC-State`. Three experiments, including both simulated and physical environments, are used to test the efficacy of our proposed algorithm. The environments are carefully chosen to demonstrate the ability of the `AC-State` agent to succeed at navigation and virtual manipulation tasks with varying degrees of difficulty; on these testbeds, algorithms with similar properties in literature fail to succeed. In what follows, we will describe the environmental setups, the function approximation scheme for the latent state, and the results that we produced.

## A.2   Mazes with Exogenous Agents and Reset Actions

We consider a global 2D maze (see Fig. 6) further divided into nine 2D maze substructures (henceforth called gridworlds). Each gridworld is made up of $6 \times 6$ ground truth states, and only one of the gridworlds contains the `AC-State` agent. Every gridworld other than the one containing the true agent has an agent placed within it whose motion is governed by random actions. An ablation where we vary the maximum number of learned codes available to the model is shown in Table 3. A table showing progress in exploration (along with the number of learned codes being used) is shown in Table 4.

| Number of Codes | Ground Truth States | % States Explored | % Dynamics Difference Error |
|---|---|---|---|
| 50 | 68 | 80.9 | 25.4 |
| 60 | 68 | 97.1 | 8.7 |
| 70 | 68 | 98.5 | 4.4 |
| 80 | 68 | 100 | 0.0 |

Table 3: Varying number of codes (learned states) available to the model while using `AC-State` on the 4-room multiple maze task. The percentage of ground states explored after 25000 samples and how the dynamics error varies as we increase the number of codes. `AC-State` typically needs more states than are actually present to fully succeed, yet performance degrades gracefully if the number of learned states is too restricted.

### A.2.1   Exploring in Presence of Reset Actions

**Data Collection:**   We collect data under a random roll-out policy while interacting with the gridworld's environment. We endow the agent with the ability to "reset" its action to a fixed starting state. The goal of this experiment is to show that, in the presence of reset actions, it is sufficiently hard for a random rollout policy to get full coverage of the mazes. To achieve sufficient coverage, we can leverage the discovered control-endogenous latent states to learn a goal seeking policy that can be incentivized to deterministically reach unseen regions of the state space. The counts of the discrete latent states are used to construct a simple tabular MDP where planning is done to reach goal states using a monte carlo version of Dijkstra's algorithm (to account for stochastic transition dynamics). Experimental results demonstrate that a goal-seeking policy achieves perfect coverage of the state space by using discovered latents for exploration, while a random policy fails to reach more than 25% of the state space in the presence of reset actions. We demonstrate this with heatmaps showing state visitation frequencies.

| Number of Samples | Ground Truth States Explored | Number of Learned States Used |
|---|---|---|
| 2000 | 27 | 36 |
| 4000 | 47 | 57 |
| 6000 | 55 | 65 |
| 8000 | 63 | 70 |
| 10000 | 63 | 72 |
| 12000 | 66 | 75 |
| 14000 | 66 | 77 |
| 16000 | 66 | 77 |
| 18000 | 68 | 79 |
| 20000 | 68 | 77 |

Table 4: Using the AC-State algorithm on the 4-room maze exploration task, we show how the number of seen ground truth states and the number of learned codes change, as we increase the number of samples seen during the exploration process.

**Experiment Details:** The encoder receives observations of size $80 \times 720 \times 3$ due to the observations from 8 other exogenous agents. The agent has an action space of 4, where actions are picked randomly from a uniform policy. For the reset action setting, we use an additional 4 reset actions, and uniformly picking a reset action can reset it to a deterministic starting state. The observation is encoded using the MLP-Mixer architecture (Tolstikhin et al., 2021) with gated residual connections (Jang et al., 2017). The model is trained using the Adam optimizer (Diederik et al., 2014) with a default learning rate of 0.0001 and without weight decay. We use a 2-layer feed-forward network (FFN) with 512 hidden units for the encoder network, followed by a vector quantization (VQ-VAE) bottleneck. The use of a VQ-VAE bottleneck would discretize the representation from the multi-step inverse model by adding a codebook of discrete learnable codes. For recovering control-endogenous latents from the maze we want to control while ignoring the other exogenous mazes, we further use a MLP-Mixer architecture Tolstikhin et al. (2021) with gated residual updates Jang et al. (2017). Both the inverse mode and the VQ-VAE bottlenecks are updated using an Adam optimizer Diederik et al. (2014) with a default learning rate of 0.0001 without weight decay.

### A.3 Robotic Arm under Exogenous Observations

Using a robotic arm (Annin, 2022) with 6 degrees of freedom, there are 5 possible abstract actions: forward, reverse, left, right, and stay. The robot arm moves within 9 possible positions in a virtual 3x3 grid, with walls between some cells. The center of each cell is equidistant from adjoining cells. The end effector is kept at a constant height. We compute each cell's centroid and compose a transformation from the joint space of the robot to particular grid cells via standard inverse kinematics calculations. Two cameras are used to take still images. One camera is facing the front of the robot, and the other camera is facing down from above the robot. When a command is received, the robot moves from one cell center to another cell center, assuming no wall is present. After each movement, still images (640x480) are taken from two cameras and appended together into one image (1280x480). During training, only the forward facing, down-sampled (256x256) image is used. Each movement takes one second. After every 500 joint space movements, we re-calibrate the robot to the grid to avoid position drift.

We collected 6 hours (14000 data samples) of the robot arm following a uniformly random policy. There were no episodes or state resets. In addition to the robot, there are several distracting elements in the image. A looped video (https://www.youtube.com/watch?v=zRpazyH1WzI) plays on a large display in high resolution (4K video) at 2x speed. Four drinking toy birds, a color-changing lamp, and flashing streamer lights are also present. During the last half hour of image collection, the distracting elements are moved and/or removed to simulate additional uncertainty in the environment. An illustration of the setup is in Figure 9, along with the specific counter-example for one-step inverse models. The quantitative performance of various baselines and `AC-State` on this task is shown in Table 5.

Figure 8: For four different baseline models we show input image (left), top-down image (center), and reconstruction of the image from the learned latent state (right). Two consecutive frames are shown for each method. Both the one-step inverse model and `AC-State` successfully discard the background distractors, but only `AC-State` does this while also successfully capturing the true position of the robot arm.

**Additional Result :** Videos of the latent state visualization for baselines and *AC-State* are shown in Figure 8. In each video, the frontal view (ground truth) is shown on the left, the top-down view (ground truth) is shown in the middle, the reconstruction of the frontal view from the latent state is shown on the right.

| Algorithms | Image Reconstruction Error | Ground Truth State Accuracy | Dynamics Difference Error (%) |
|---|---|---|---|
| Contrastive | 45.20 | 33.05 | 66.1 |
| Autoencoder | 50.56 | 20.82 | 82.2 |
| Forward Generative | 65.48 | 21.30 | 76.4 |
| 1-Step Inverse | 88.70 | 86.75 | 16.6 |
| AC-State | 88.04 | 99.83 | 0.52 |

Table 5: Robot arm results with various baselines. We show error in reconstructing the original image (generally higher is better as it indicates that exogenous noise is discarded from the latent state). We also train a probe network to predict the ground truth endogenous state from the learned latent state as well as dynamics difference error from the learned latent dynamics.

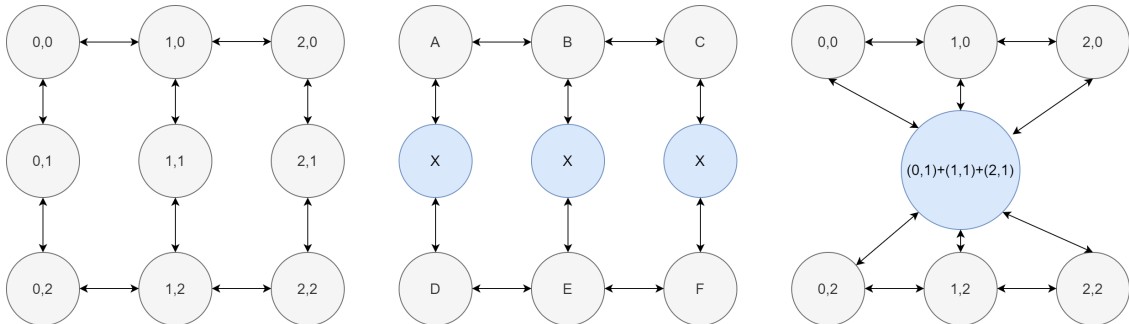

Figure 9: The robot arm has five actions and moves within nine possible control-endogenous states (left). The transition directions are indicated by the arrows. For example, if the robot arm is at (0,0) and selects the down action, it moves to (0,1), but if it selects the up action, it remains at the same position. A simple inverse model can achieve perfect accuracy even if the middle row of true control-endogenous states are mapped to a single latent state (middle). This leads the one-step inverse model to merge them (right).

### A.4 Matterport Simulator with Exogenous Observations

We evaluated *AC-State* on the matterport simulator introduced in Chang et al. (2017). The simulator contains indoor houses in which an agent can navigate. The house contains a finite number of viewpoints which the agent can navigate to. At each viewpoint, the agent has control of its viewing angle (by turning left or right by an angle) and its elevation: in total there are 12 possible viewing angles per viewpoint and 3 possible elevations. We collect data using a random rollout policy. At each step of the rollout policy, the agent navigates to a neighbouring viewpoint. We also randomly change the agent elevation at some of steps of the rollout policy, in order to introduce exogenous information which the agent cannot control. We collect a single long episode of 20,000 state-transitions. The control-endogenous latent state in this setup is the viewpoint information while the exogenous information is the information regarding agent elevation.

**Experimental Details:** The model input is the panorama of the current viewpoint i.e. 12 images for the 12 possible views of each viewpoint. The `AC-State` model $f$ is parameterized using a vision transformer (ViT) Dosovitskiy et al. (2021). Each view within the panorama is fed separately into the ViT as a sequence of patches along with a learnable token called the class (or CLS) following the procedure in Dosovitskiy et al. (2021). To obtain the viewpoint representation, we take the representation corresponding to the CLS token of each view and take the mean across all views. We discretize this representation using a VQ-VAE bottleneck van den Oord et al. (2017) to obtain the final representation. We use a 6-layer transformer with 256 dimensions in the embedding. We use a feedforward network (FFN) after every attention operation in the ViT similar to Vaswani et al. (2017). The FFN is a 2 layer MLP with a GELU activation Hendrycks & Gimpel (2016) which first projects the input to a higher dimension $D$ and then projects it back to the original dimension. We set the FFN dimension $D$ to 512. We use 4 heads in the ViT. We train the model for 20 epochs using Adam optimizer Diederik et al. (2014) with learning rate 1e-4. The model is trained to predict the viewpoint of the next state as the action.

**Baselines** We use 3 baselines for comparison - (1) Auto-Encoder, (2) Generative, and (3) Contrastive. **Auto-Encoder** - In this baseline, we feed the observation as input to a ViT encoder and a ViT decoder is trained to reconstruct the observation. **Generative** - In this baseline, the model is trained to predict the next observation given the current observation and the action. The encoder and decoder both are implemented using a ViT model. **Contrastive** - In this baseline, we use the simclr objective (Chen et al., 2020). We first pass the observation through a ViT encoder. We take the representation corresponding to the CLS token and split into two portions. We then apply the simclr objective considering the two representations from the same observation as the positive pairs and other combinations as the negative pairs. We also tried a temporal contrastive objective instead of simclr, but found that simclr had better performance in ignoring exogenous noise.

**Results:** We present the results for this experiment in Figure 7 (right). The *Controllable Latent State Accuracy* is the viewpoint prediction accuracy for the current state. The *Exogenous Noise-Ignoring Accuracy.* is calculate as $1 - \frac{\mathcal{E}-33.33}{100-33.33}$, where $\mathcal{E}$ is elevation prediction accuracy. Thus a higher elevation prediction accuracy leads to a lower the exogenous noise-inducing accuracy. We can see that the proposed *AC-State* model has the highest control-endogenous latent state and exogenous noise-ignoring accuracy. Thus, it outperforms the baselines we considered at capturing Control-endogenous Latent State information while ignoring exogenous noise. We calculated state parsimony as $\frac{\text{Num. Ground Truth States}}{\text{Num. Discovered States}}$. Therefore, a lower state parsimony denotes a high number of discovered states which means that the model fails at ignoring exogenous information. The proposed model has the highest state parsimony which shows the effectiveness of the model in ignoring the exogenous noise whilst only capturing control-endogenous latent state. Detailed results with mean and standard deviation over three seeds are reported in Table 6.

| Algorithms | Endogenous Accuracy | Exogenous Noise-Ignoring Accuracy (%) |
| --- | --- | --- |
| Autoencoder | $86.15 \pm 1.33$ | $40.08 \pm 1.24$ |
| Contrastive | $93.57 \pm 1.07$ | $10.05 \pm 1.19$ |
| Forward Generative | $97.25 \pm 0.58$ | $3.66 \pm 1.19$ |
| AC-State (No Bottleneck) | $96.94 \pm 0.58$ | $47.81 \pm 2.77$ |
| AC-State | $\underline{98.90 \pm 0.53}$ | $\underline{91.34 \pm 1.46}$ |

Table 6: Experimental Results with Matterport with mean and standard deviation over three seeds.

## B    Additional Discussion on Related Works

In this section, we provide a clear example of why an one-step inverse model might not be useful to ignore exogenous noise. The idea of using a simple one step inverse dynamics models have been explored in the past (Pathak et al., 2017; Bharadhwaj et al., 2022), yet the one step inverse model has counterexamples establishing that it fails to capture the full control-endogenous latent state (Efroni et al., 2022c; Misra et al., 2020; Rakelly et al., 2021; Hutter & Hansen, 2022). Intuitively, the 1-step inverse model is under-constrained and thus may incorrectly merge distinct states which are far apart in the MDP but have a similar local structure. As a simple example, suppose we have a cycle of states: $s_1, s_2, s_3, s_4, s_5, s_6$ where $a = 0$ moves earlier in the cycle and $a = 1$ moves later in the cycle. Suppose $s_1, s_4$ are merged into a distinct latent state $s_i$, $s_2, s_5$ are merged into a distinct latent state $s_j$ and $s_3, s_6$ are merged into a distinct latent state $s_k$. The inverse-model examples are: $(s_i, s_j, 1), (s_j, s_i, 0), (s_j, s_k, 1), (s_k, s_j, 0), (s_k, s_i, 1), (s_i, s_k, 0)$. Because all of these examples have distinct inputs, a 1-step inverse model still has zero error despite the incorrect merger of $\{s_1, s_4\}$, $\{s_2, s_5\}$, and $\{s_3, s_6\}$.

## C    Algorithm Details

We provide more detailed algorithm descriptions for `AC-State` with a planning policy (Algorithm 2). The latter uses a findgoal function which selects a low-count state using breadth-first search along with a plan function that finds an optimal action using Dijkstra's algorithm.

---

**Algorithm 2** AC-State for Latent State Discovery using a Planning Policy

---

1: Initialize a replay buffer $D$. Initialize encoder $f_\theta$. Assume a number of samples to collect $T$, a set of actions $\mathcal{A}$.
2: $x_1 \sim U(\mu(x))$, $a_1 \sim U(\mathcal{A})$, $t_g := 1$, and $\mathcal{T} = \{\}$
3: **for** t = 1, 2, ..., $T$ **do**
4:     $x_{t+1} \sim \mathbb{P}(x'|x_t, a_t)$
5:     $s_t = f_\theta(x_t)$
6:     $s_{t+1} = f_\theta(x_{t+1})$
7:     Update tabular-MDP $\mathcal{M}$ with $(s_t, a_t, s_{t+1})$.
8:     **if** $t = t_g$ **then**
9:         Pick a new goal
10:         $t_s := t$
11:         $t_g, g := findgoal(s_t, \mathcal{M})$
12:         $a_t \sim U(\mathcal{A})$
13:         Add $t$ to $\mathcal{T}$
14:     **else**
15:         $a_t := plan(s_t, g, \mathcal{M})$
16:     $K_t := t_g - t$
17:     $j \sim \mathcal{T}$ and $k \sim U(j, K_j)$
18:     $\mathcal{L} = \mathcal{L}_{\texttt{AC-State}}(f_\theta, j, x, a, k) + \mathcal{L}_{\text{Bottleneck}}(f_\theta, x_j) + \mathcal{L}_{\text{Bottleneck}}(f_\theta, x_{j+k})$
19:     Update $\theta$ to minimize $\mathcal{L}$ by gradient descent.

---

# D   Detailed Theory and Discussion

## D.1   High-Level Overview of Theory

We present an asymptotic analysis of `AC-State` showing it recovers the control-endogenous latent state encoder $f_\star$. The mathematical model we consider is the deterministic Ex-BMDP. There, the transition model of the latent state decomposes into a control-endogenous latent state, which evolves deterministically, along with a noise term–the uncontrol-endogenous portion of the state. The noise term may be an arbitrary temporally correlated stochastic process. If the reward does not depend on this noise, any optimal policy may be expressed in terms of this control-endogenous latent state. In this sense, the recovered control-endogenous latent state is sufficient for achieving optimal behavior.

Intuitively, the Ex-BMDP is similar to a video game, in which a "game engine" takes player actions and keeps track of an internal game state (the control-endogenous state component), while the visuals and sound are rendered as a function of this compact game state. A modern video game's core state is often orders of magnitude smaller than the overall game.

The algorithm we propose for recovering the optimal control-endogenous latent state involves ($i$) an action prediction term; and ($ii$) a mutual information minimization term. The action prediction term forces the learned representation $\widehat{f}(x)$ to capture information about the dynamics of the system. At the same time, this representation for $\widehat{f}(x)$ (which is optimal for action-prediction) may also capture information which is unnecessary for control. In our analysis we assume that $\widehat{f}(x)$ has discrete values and show the control-endogenous latent state is the unique coarsest solution.

To enable more widespread adoption in deep learning applications, we can generalize this notion of coarseness to minimizing mutual information between $x$ and $f(x)$. These are related by the data-processing inequality; coarser representation reduces mutual information with the input. Similarly, the notion of mutual information is general as it does not require discrete representation.

## D.2 The Control-Endogenous Partition is a Bayes' Optimal Solution

Consider the generative process in which $x$ is sampled from a distribution $\mu$, the agent executes a policy $\pi$ for $t$ time steps and samples $x'$. Denote by $\mathbb{P}_{\pi,\mu}(x, x', t)$ as the joint probability, and by $\mathbb{P}_{\pi,\mu}(a \mid x, x', t)$ as the probability that under this generative process the action upon observing $x$ is $a$. The following proof for Proposition 5.1, which builds on Proposition 3.3, shows that the optimal bayes solution $\mathbb{P}_{\pi,\mu}(a \mid x, x', t)$ is equal to $\mathbb{P}_{\pi,\mu}(a \mid f_\star(x), f_\star(x'), t)$ for $\mathbb{P}_{\pi,\mu}(x, x', t) > 0$, where $\mathbb{P}_{\pi,\mu}(x, x', t) > 0$ is the probability to sample $x$ .

**Proposition 5.1.** *Assume that $\pi$ is an endogenous policy. Let $x \sim \mu$ for some distribution $\mu$. Then, the Bayes' optimal predictor of the action-prediction model is piece-wise constant with respect to the control-endogenous partition: for all $a \in \mathcal{A}$, $t > 0$ and $x, x' \in \mathcal{X}$ such that $\mathbb{P}_{\pi,\mu}(x, x', t) > 0$ it holds that*

$$\mathbb{P}_{\pi,\mu}(a \mid x, x', t) = \mathbb{P}_{\pi,\mu}(a \mid f_\star(x), f_\star(x'), t).$$

*Proof.* Assume that $\pi$ is and endogenous policy. Let $x \sim \mu$ for some distribution $\mu$. Then, the Bayes' optimal predictor of the action-prediction model is piece-wise constant with respect to the control-endogenous partition: for all $a \in \mathcal{A}$, $t > 0$ and $x, x' \in \mathcal{X}$ such that $\mathbb{P}_{\pi,\mu}(x, x', t) > 0$ it holds that:

$$\mathbb{P}_{\pi,\mu}(a \mid x, x', t) = \mathbb{P}_{\pi,\mu}(a \mid f_\star(x), f_\star(x'), t).$$

$\square$

We comment that the condition $\mathbb{P}_{\pi,\mu}(x, x', t) > 0$ is necessary since ,otherwise, the conditional probability $\mathbb{P}_{\pi,\mu}(a \mid x, x', t)$ is well not defined.

Proposition 5.1 is readily proved via the factorization of the future observation distribution to control-endogenous and exogenous parts that holds when the executed policy does not depend on the exogenous state (Proposition 3.3).

*Proof.* The proof follows by applying Bayes' theorem, Proposition equation 3.3, and eliminating terms from the numerator and denominator.

Fix any $t > 0$, $x, x' \in \mathcal{X}$ and $a \in \mathcal{A}$ such that $\mathbb{P}_{\pi,\mu}(x', x, t) > 0$. Let $s = f_\star(x)$, $s' = f_\star(x')$, $e = f_{\star,e}(x)$, and $e' = f_{\star,e}(x')$. The following relations hold.

$$
\begin{aligned}
&\mathbb{P}_{\pi,\mu}(a \mid x', x, t) \\
&\overset{(a)}{=} \frac{\mathbb{P}_{\pi,\mu}(x' \mid x, a, t)\mathbb{P}_{\pi,\mu}(a \mid x)}{\sum_{a'} \mathbb{P}_{\pi,\mu}(x' \mid x, a', t)} \\
&\overset{(b)}{=} \frac{\mathbb{P}_{\pi,\mu}(x' \mid x, a, t)\pi(a \mid s)}{\sum_{a'} \mathbb{P}_{\pi,\mu}(x' \mid x, a', t)\pi(a' \mid s)} \\
&\overset{(c)}{=} \frac{q(x' \mid s', e')\mathbb{P}_{\pi,\mu}(s' \mid s, a, t)\mathbb{P}_{\pi,\mu}(e' \mid e, t)\pi(a \mid s)}{\sum_{a'} q(x' \mid s', e')\mathbb{P}_{\pi,\mu}(s' \mid s, a', t)\mathbb{P}_{\pi,\mu}(e' \mid e, t)\pi(a' \mid s)} \\
&= \frac{\mathbb{P}_{\pi,\mu}(s' \mid s, a, t)\pi(a \mid s)}{\sum_{a'} \mathbb{P}_{\pi,\mu}(s' \mid s, a', t)\pi(a' \mid s)}.
\end{aligned}
$$

Relation $(a)$ holds by Bayes' theorem. Relation $(b)$ holds by the assumption that $\pi$ is endogenous. Relation $(c)$ holds by Proposition equation 3.3.

Thus, $\mathbb{P}_{\pi,\mu}(a \mid x', x, t) = \mathbb{P}_{\pi,\mu}(a \mid f_\star(x'), f_\star(x), t)$, and is constant upon changing the observation while fixing the control-endogenous latent state. $\square$

## D.3 Stationary Assumption

We make the following assumptions on the policy by which the data is collected.

**Assumption D.1.** *Let $T_{\mathcal{D}}(s' \mid s)$ be the Markov chain induced on the control-endogenous state space by executing the policy $\pi_{\mathcal{D}}$ by which AC-State collects the data.*

1. *The Markov chain $T_{\mathcal{D}}$ has a stationary distribution $\mu_{\mathcal{D}}$ such that $\mu_{\mathcal{D}}(s, a) > 0$ and $\pi_{\mathcal{D}}(a \mid s) \geq \pi_{\min}$ for all $s \in \mathcal{S}$ and $a \in \mathcal{A}$.*

2. *The policy $\pi_{\mathcal{D}}$ by which the data is collected reaches all accessible states from any states. For any $s, s' \in \mathcal{S}$ and any $h > 0$ if $s'$ is reachable from $s$ then $\mathbb{P}_{\mathcal{D}}(s' \mid s, h) > 0$.*

3. *The policy $\pi_{\mathcal{D}}$ does not depend on the exogenous state, and is an endogenous policy.*

See Levin & Peres (2017), Chapter 1, for further discussion on the classes of Markov chains for which the assumption on the stationary distribution hold. The second and third assumptions are satisfied for the the random policy that simply executes random actions.

We consider the stochastic process in which an observation is sampled from a distribution $\mu$ such that $\mu(s) = \mu_{\mathcal{D}}(s)$ for all $s$. Then, the agent executes the policy $\pi_{\mathcal{D}}$ for $t$ time steps. For brevity, we denote the probability measure induced by this process as $\mathbb{P}_{\mathcal{D}}$.

### D.4   The Coarsest Partition is the Control-Endogenous State Partition

Proposition 5.1 from previous section shows that the multi-step action-prediction model is piece-wise constant with respect to the partition induced by the control-endogenous states $f_{\star} : \mathcal{X} \to [S]$. In this section, we assume that the executed policy is an endogenous policy that induces sufficient exploration. With this, we prove that there is no coarser partition of the observation space such that the set of inverse models are piece-wise constant with respect to it.

We begin by defining several useful notions. We denote the set of reachable control-endogenous states from $s$ in $h$ time steps as $\mathcal{R}_h(s)$.

**Definition 5.2.** (Reachable Control-endogenous States).  *Let the set of reachable control-endogenous states from $s \in \mathcal{S}$ in $h > 0$ time steps be $\mathcal{R}(s, h) = \{s' \mid \max_{\pi} \mathbb{P}_{\pi}(s' \mid s, h) = 1\}$.*

Observe that every reachable state from $s$ in $h$ time steps satisfies that $\max_{\pi} \mathbb{P}_{\pi,\mu}(s' \mid s_0 = s, h) = 1$ due to the deterministic assumption of the control-endogenous dynamics.

Next, we define a notion of *consistent partition* with respect to a set of function values. Intuitively, a partition of space $\mathcal{X}$ is consistent with a set of function values if the function is piece-wise constant on that partition.

**Definition 5.3.** (Consistent Partition with respect to $\mathcal{G}$).  *Consider a set $\mathcal{G} = \{g(a, y, y')\}_{y,y' \in \mathcal{Y}, a \in \mathcal{A}}$ where $g : \mathcal{A} \times \mathcal{Y} \times \mathcal{Y} \to [0, 1]$. We say that $f : \mathcal{Y} \to [N]$ is a* consistent partition *with respect to $\mathcal{G}$ if for all $y, y_1', y_2' \in \mathcal{Y}$, $f(y_1') = f(y_2')$ implies that $g(a, y, y_1') = g(a, y, y_2')$ for all $a \in \mathcal{A}$.*

Observe that Proposition 5.1 shows that the partition of $\mathcal{X}$ according to $f_{\star}$ is consistent with respect to $\{\mathbb{P}_{\mathcal{D}}(a \mid x, x', h) \mid x, x' \in \mathcal{X}, h \in [H] \text{ s.t. } \mathbb{P}_{\mathcal{D}}(x, x', h) > 0\}$, since, by Proposition 5.1, $\mathbb{P}_{\mathcal{D}}(a \mid x, x', h) = \mathbb{P}_{\mathcal{D}}(a \mid f_{\star}(x), f_{\star}(x'), h)$.

Towards establishing that the coarsest abstraction according to the `AC-State` objective is $f_{\star}$ we make the following definition.

**Definition 5.4.** (The Generalized Inverse Dynamics Set $\mathrm{AC}(s, h)$).  *Let $s \in \mathcal{S}, h \in \mathbb{N}$. We denote by $\mathrm{AC}(s, h)$ as the set of multi-step inverse models accessible from $s$ in $h$ time steps. Formally,*

$$\mathrm{AC}(s, h) = \{\mathbb{P}_{\mathcal{D}}(a \mid s', s'', h') : s' \in \mathcal{R}(s, h - h'), s'' \in \mathcal{R}(s', h'), a \in \mathcal{A}, h' \in [h]\}. \tag{5}$$

Observe that in equation equation 5 the inverse function $\mathbb{P}_{\mathcal{D}}(a \mid s', s'', h')$ is always well defined since $\mathbb{P}_{\mathcal{D}}(s', s'', h') > 0$. It holds that

$$\mathbb{P}_{\mathcal{D}}(s', s'', h') = \mathbb{P}_{\mathcal{D}}(s')\mathbb{P}_{\mathcal{D}}(s'' \mid s', h') > 0,$$

since $\mathbb{P}_{\mathcal{D}}(s') > 0$ and $\mathbb{P}_{\mathcal{D}}(s'' \mid s', h') > 0$. The inequality $\mathbb{P}_{\mathcal{D}}(s') > 0$ holds by the assumption that the stationary distribution when following $\mathcal{U}$ has positive support on all control-endogenous states (Assumption D.1). The inequality $\mathbb{P}_{\mathcal{D}}(s'' \mid s', h') > 0$ holds since, by definition $s'' \in \mathcal{R}(s, h')$ is reachable from $s'$ in $h'$ time steps; hence, $\mathbb{P}_{\mathcal{D}}(s'' \mid s', h') > 0$ by the fact that Assumption D.1 implies that the policy $\pi_{\mathcal{D}}$ induces sufficient exploration.

**Theorem 5.5.** ($f_\star$ is the coarsest partition consistent with respect to *AC-State* objective). *Given the bounded diameter Assumption 3.2 and the high-coverage policy assumption (Appendix D.3) hold, it follows that there is no coarser partition than $f_\star$, which is consistent with* $\mathrm{AC}(s, D)$ *for any* $s \in \mathcal{S}$.

Assume 3.2 and D.1 holds. Then there is no coarser partition than $f_\star$ which is consistent with $\mathrm{AC}(s, D)$ for any $s \in \mathcal{S}$.

*Proof.* We will show inductively that for any $h > 0$ and $s \in \mathcal{S}$ there is no coarser partition than $\mathcal{R}(s, h)$ for the set $\mathcal{R}(s, h)$ that is consistent with $\mathrm{AC}(s, h)$. Since the set of reachable states in $h = D$ time steps is $\mathcal{S}$–all states are reachable from any state in $D$ time steps–it will directly imply that there is no coarser partition than $\mathcal{R}(s, D) = \mathcal{S}$ consistent $\mathrm{AC}(s, D)$.

**Base case, $h = 1$.** Assume that $h = 1$ and fix some $s \in \mathcal{S}$. Since the control-endogenous dynamics is deterministic, there are $A$ reachable states from $s$. Observe the inverse dynamics for any $s' \in \mathcal{R}(s, 1)$ satisfies that

$$\mathbb{P}_{\mathcal{D}}(a \mid s, s', 1) = \begin{cases} 1 & \text{if } a \text{ leads } s' \text{ from } s \\ 0 & \text{o.w.} \end{cases}. \tag{6}$$

This can be proved by an application of Bayes' rule:

$$\mathbb{P}_{\mathcal{D}}(a \mid s, s', 1) = \frac{\mathbb{P}_{\mathcal{D}}(s' \mid s = s, a, 1)\pi_{\mathcal{D}}(a \mid s)}{\sum_{a'} \mathbb{P}_{\mathcal{D}}(s' \mid s = s, a', 1)\pi_{\mathcal{D}}(a' \mid s)}$$
$$= \frac{T(s' \mid s, a)\pi_{\mathcal{D}}(a \mid s)}{\sum_{a'} T(s' \mid s, a')\pi_{\mathcal{D}}(a' \mid s)}$$
$$\begin{cases} \geq \pi_{\min} & (s, a) \text{ leads to } s' \\ = 0 & \text{o.w.} \end{cases},$$

where the last relation holds by Assumption D.1. Furthermore, observe that since $s' \in \mathcal{R}(s, 1)$, i.e., it is reachable from $s$, the probability function $\mathbb{P}_{\mathcal{D}}(a \mid s, s', 1)$ is well defined.

Hence, by equation equation 6, we get that for any $s'_1, s'_2 \in \mathcal{R}(s, 1)$ such that $s'_1 \neq s'_2$ it holds that exists $a \in \mathcal{A}$ such that

$$\pi_{\min} \geq \mathbb{P}_{\mathcal{D}}(a \mid s, s'_1, 1) \neq \mathbb{P}_{\mathcal{D}}(a \mid s, s'_2, 1) = 0.$$

Specifically, choose $a$ such that taking $a$ from $s$ leads to $s'_1$ and see that, by equation equation 6,

$$\pi_{\min} \geq \mathbb{P}_{\mathcal{D}}(a \mid s, s'_1, 1) \neq \mathbb{P}_{\mathcal{D}}(a \mid s, s'_2, 1) = 0.$$

Lastly, by the fact that $s \in \mathcal{S}$ is an arbitrary state, the induction base case is proved for all $s \in \mathcal{S}$.

**Induction step.** Assume the induction claim holds for all $t \in [h]$ where $h \in \mathbb{N}$. We now prove it holds for $t = h + 1$.

Fix some $s \in \mathcal{S}$. We prove the induction step and show that $\mathcal{R}(s, h+1)$ is the coarsest partition which is consistent $\mathrm{AC}(s, h+1)$. Meaning, there exists $\tilde{s}, t \in [h+1], a$ such that $\mathbb{P}(a \mid \tilde{s}, s'_1, 1), \mathbb{P}(a \mid \tilde{s}, s'_1, 1) \in \mathrm{AC}(\bar{s}, h+1)$ and

$$\mathbb{P}_{\mathcal{D}}(a \mid \tilde{s}, s'_1, t) \neq \mathbb{P}_{\mathcal{D}}(a \mid \tilde{s}, s'_2, t). \tag{7}$$

Observe that, by Definition 5.4, it holds that,

$$\mathrm{AC}(s, h+1) = \{\mathbb{P}_{\mathcal{D}}(a \mid s, s', h+1)\}_{s' \in \mathcal{R}(s, h+1)} \cup_{\bar{s} \in \mathcal{R}(s, 1)} \mathrm{AC}(\bar{s}, h).$$

Meaning, the set $\mathrm{AC}(s, h+1)$ can be written as the union of (1) the set $\{\mathbb{P}_\mathcal{D}(a \mid s, s', h+1)\}_{s' \in \mathcal{R}(s, h+1)}$, and (2) the union of the sets $\mathrm{AC}(\bar{s}, h)$ for all $\bar{s}$ which is reachable from $s$ in a single time step.

By the induction hypothesis, the coarsest partition which is consistent with $\mathrm{AC}(\bar{s}, h)$ is $\cup_{h'=1}^h \mathcal{R}(\bar{s}, h')$. We only need to prove, that for any $\bar{s}_1, \bar{s}_2 \in \mathcal{R}(s, h+1)$ such that $\bar{s}_1 \neq \bar{s}_2$ exists some $a \in \mathcal{A}, h' \in [h]$ and $s_{h'} \in \mathcal{R}(s, h+1)$ such that

$$\mathbb{P}_\mathcal{D}(a \mid s_{h'}, \bar{s}_1, h') \neq \mathbb{P}_\mathcal{D}(a \mid s_{h'}, \bar{s}_2, h'),$$

this will imply that the set of reachable states in $h + 1$ time states is also the coarsest partition which is consistent with $\mathrm{AC}(s, h+1)$.

Fix $\bar{s}_1, \bar{s}_2 \in \mathcal{R}(s, h+1)$ such that $\bar{s}_1 \neq \bar{s}_2$ we show that exists a certificate in $\mathrm{AC}(s, h+1)$ that differentiate between the two by considering three cases.

1. **Case 1:** Both $\bar{s}_1$ and $\bar{s}_2$ are reachable from all $s' \in \mathcal{R}(s, 1)$. In this case, for all $s' \in \mathcal{R}(s, 1)$ it holds that $\bar{s}_1, \bar{s}_2 \in \mathcal{R}(s', h)$. By the induction hypothesis, $\bar{s}_1$ and $\bar{s}_2$ cannot be merged while being consistent with $\mathrm{AC}(s', h)$.

2. **Case 2:** Exists $s' \in \mathcal{R}(s, 1)$ such $\bar{s}_1$ is reachable from $s'$ in $h$ time steps and $\bar{s}_2$ is not. Let $a$ be the action that leads to $s'$ from state $s$. In that case, it holds by the third assumption of Assumption D.1 that

$$
\begin{aligned}
\mathbb{P}_\mathcal{D}(a \mid s, \bar{s}_1, h+1) &\overset{(a)}{=} \frac{\mathbb{P}_\mathcal{D}(\bar{s}_1 \mid s, a, h+1)\pi_\mathcal{D}(a \mid s)}{\sum_{a'} \mathbb{P}_\mathcal{D}(\bar{s}_1 \mid s, a', h+1)\pi_\mathcal{D}(a' \mid s)} \\
&\overset{(b)}{=} \frac{\mathbb{P}_\mathcal{D}(\bar{s}_1 \mid s', h)\pi_\mathcal{D}(a \mid s)}{\sum_{a'} \mathbb{P}_\mathcal{D}(\bar{s}_1 \mid s, a', h+1)\pi_\mathcal{D}(a' \mid s)} \\
&\overset{(c)}{\geq} \pi_{\min} \frac{\mathbb{P}_\mathcal{D}(\bar{s}_1 \mid s', h)}{\sum_{a'} \mathbb{P}_\mathcal{D}(\bar{s}_1 \mid s, a', h+1)} \\
&\overset{(d)}{>} 0.
\end{aligned}
\tag{8}
$$

Relation (a) holds by Bayes' rule. Relation (b) holds by the fact that $(s, a)$ determinstically leads to $s'$. Relation (c) and (d) holds by Assumption D.1.

Observe that $\mathbb{P}_\mathcal{D}(a \mid s, \bar{s}_2, h+1) = 0$ since $\bar{s}_2$ is not reachable upon taking action $a$ from state $s$, by the assumption. Combining this fact with equation equation 8 implies that

$$0 < \mathbb{P}_\mathcal{D}(a \mid s, \bar{s}_1, h+1) \neq \mathbb{P}_\mathcal{D}(a \mid s, \bar{s}_2, h+1) = 0.$$

Hence, exists a certificate that differentiates between $\bar{s}_1$ and $\bar{s}_2$. Observe that since $\bar{s}_2 \in \mathcal{R}(s, h+1)$, i.e., it is reachable from $s$, it holds that $\mathbb{P}_\mathcal{D}(a \mid s, \bar{s}_2, h+1) = 0$, i.e., it is well defined.

3. **Case 3:** Exists $s' \in \mathcal{R}(s, 1)$ such $\bar{s}_2$ is reachable from $s'$ in $h$ time steps and $\bar{s}_1$ is not. Symmetric to case 2.

This establishes the result we needed to show in equation equation 7 and, hence, the induction and result hold. $\qquad\square$

## E  Discussion: What does Control-Endogenous Latent State Capture? Does it ignore Task-Relevant and Reward-Relevant Information?

The control-endogenous latent state is defined by having factorized dynamics that depend on actions, while the exogenous dynamics do not depend on actions. If we refer to the control-endogenous state as $s$ and the exogenous state as $e$, we can write the factorized dynamics as: $\mathbb{P}(s'|s, a)\mathbb{P}(e'|e)$. In particular, we want to find the smallest $s$ such that the latent dynamics follow this factorization.

Causal Dynamics Learning (Wang et al., 2022b) provided some definitions that are helpful for building intuitions about the semantics of the control-endogenous state. In their work, they decompose a factorized state into controllable factors, action-relevant factors, and action-irrelevant factors. They define controllable factors as any causal descendants of actions or other controllable factors. They define action-relevant factors as causal parents of controllable factors or other action-relevant factors. All other factors are defined as action-irrelevant. While `AC-State` learns a general encoder instead of relying on a given factorization, the special case of known factors can be used to build intuition.

An example is given in Wang et al. (2022b) involving six known factors: $z_1, z_2, z_3, z_4, z_5, z_6$. The dynamics are factorized as: $\mathbb{P}(z_1'|z_1, a)\mathbb{P}(z_2'|z_1, z_2, z_3)\mathbb{P}(z_3'|z_3, z_4)\mathbb{P}(z_4'|z_4)\mathbb{P}(z_5'|z_5, z_6)\mathbb{P}(z_6'|z_6)$. We can assign these factors to either the control-endogenous state or exogenous state. The smallest way to define the control-endogenous state is: $s = (z_1, z_2, z_3, z_4)$, while the exogenous state is set to $e = (z_5, z_6)$. Intriguingly, the control-endogenous state includes many factors which are not controllable.

As a simple example, we can imagine a robotic arm that is allowed to interact with two blocks (one red and one blue). What is the control-endogenous latent state for this environment? It clearly includes the robotic arm itself, as it can be manipulated using the actions. Likewise, the physical properties of the blocks (their position and weight) will be included in the control-endogenous latent state.

The color of the blocks is not part of the control-endogenous latent state. What if our task is to retrieve the red block? In general, the control-endogenous latent state will not be sufficient for solving the task. If task completion has a causal effect on the rest of the control-endogenous state, then the task relevant information is control-endogenous. As a concrete example, if the agent were forced to redo the task of retrieving the red block until successful, then the color of the blocks would become control-endogenous. Intuitively, in the most natural and realistic environments, rewards and task completion will have a causal effect on the rest of the control-endogenous state. On the other hand, if the episodes are terminated immediately after a reward is received, it would be useful to add reward prediction as an auxiliary task for learning the representation.

### E.1 AC-State Captures the Position of a Block that is Movable by the Agent

We also ran experiments to verify that `AC-State` performs correctly in an environment with an agent and a movable block. According to theory, both the agent and the movable block should be captured by the representation. If we write the agent's position as $s_A$ and the block's position as $s_B$, the ground truth dynamics are $\mathbb{P}(s_A', s_B'|s_A, s_B, a) = \mathbb{P}(s_A'|s_A, a)\mathbb{P}(s_B'|s_A, s_B, a)$. Note the asymmetry: the agent affects the block but the block does not affect the agent. We tested two variants of this setup where the agent either pulls or pushes the block. In the pull environment, the block is pulled alongside the agent unless it moves to the outer edge of the environment, in which case the agent drops the block. In the push environment, the agent moves the block if it moves into it, and the agent/block wraps around to the other side of the grid (like a torus) if it moves off the edge. These setups are designed so that the block cannot be stuck in a corner of the grid.

`AC-State` was trained on 500k samples in each environment with a uniformly random rollout policy. Probing revealed that both $s_A$ and $s_B$ were perfectly captured by the encoded representation. Planning was performed using Dijkstra's algorithm and example rollouts are included in the supplementary material as pull_examples.txt and push_examples.txt.

