# OpenReview forum: "Guaranteed Discovery of Control-Endogenous Latent States with Multi-Step Inverse Models"
_TMLR — Accepted by TMLR_

### Review · Reviewer_jojo · 2022-12-11

**Summary Of Contributions:**

This paper proposes a representation learning algorithm, AC-State, which is designed to exactly recover the a discrete control-relevant latent space (the "control-endogenous state") while discarding other control-irrelevant information (the "exogenous state"). The algorithm relies on several assumptions: a discrete latent space, a bounded diameter (e.g. no absorbing states), deterministic dynamics, and that the data generated to learn the representation is generated by a endogenous policy (i.e. which is not sensitive to the exogenous state). The method works by learning an encoder $f(x)$ which maps observations to latents and an inverse k-step model in the latent space which predicts $a_t$ given $f(x_t)$ and $f(x_{t+k})$. It also incorporates an information bottleneck to restrict the capacity of $f$. The paper presents both a theoretical analysis of AC-State as well as an empirical one, demonstrating theoretically that AC-State will converge to the true control-endogenous latent space in the limit and empirically that it does so in practice. Empirical experiments are presented in three domains: a robotic arm with visual distractors (e.g. video playing in the background), a maze with distractor agents, and navigation in the Matterport environment.

**Audience:**

Yes

**Broader Impact Concerns:**

No concerns.

**Claims And Evidence:**

No

**Requested Changes:**

The paper needs to be significantly reorganized and parts rewritten in order to present the ideas in a clearer fashion before I would be able to recommend acceptance. I have a number of specific suggestions for how this can be done, some of which I think are necessary and some of which I'd just strongly recommend.

Necessary changes:
- I did not find Figure 1 to be particularly helpful in understanding the method. It's not obvious which parts are learned or how they are learned (e.g. what the losses are). There is no mention of the information bottleneck which appears to be an important part of AC-State. The caption is also insufficient and should give a summary of the method.
- Section 3 discusses a number of related approaches which attempt to discard exogenous noise, but dismisses them because they are not theoretically guaranteed to obtain the control-endogenous state. This is certainly a weakness of these approaches, but more explanation is needed regarding how AC-State differs from these approaches at a high-level. (Ideally, at least some of these methods would be included as baselines to demonstrate the situations in which AC-State's theoretical guarantees make a difference, though this is not strictly necessary for acceptance).
- Table 1 lists several methods with no explanation, description, or citations (e.g. PPE, DBC, etc.). If they are important enough to be included in a table then they need to be named and discussed in the main text too. The table caption should also include citations for these methods.
- More explanation is needed to clarify the difference between Efroni et al. (2022c). I understand that the current paper removes an assumption about the initial control-endogenous state, but in what other ways are the papers the same/different? Is the method otherwise pretty much the same? Does it have the same limitations regarding discrete state spaces, bounded diameter, etc.? Can it be included as a baseline in the experiments later in the paper?
- Page 7 introduces the information bottleneck and states that it is important that the method finds the lowest-capacity f. This claim is not explained or justified. Please add an explanation and intuition for why this is needed.
- The loss term $\mathcal{L}_\mathrm{Bottleneck}$ is not defined.
- The overall algorithm including both loss terms and the architecture should be summarized somewhere (ideally in a figure).
- The subsections on Page 8 ("AC-State with Random Policy" and "AC-State with Planning Policy") seem like they belong in the experiments section, as they are not really parts of the core method but rather approaches for data collection. Though that said, I was confused about how "AC-State with Planning Policy" works. Is the idea that you are given a planning policy, which is just used for data gathering to train AC-State? Or that it's used on top of AC-State (in which case, I would wonder how is the latent space learned)? Or is it used in-the-loop with AC-State (in which case, are there any stability issues with training, that often occur in deep RL)?
- Section 5.1 seems to be from a previous paper. Is it necessary to reproduce here? It also feels a little bit like math for math's sake---isn't it kind of by definition that $P(a|x,x')=P(a|f(x),f(x')$ if $f(x)$ recovers the endogenous state and $a$ is determined solely by the endogenous state? Also, it seems that an important conclusion from this section is that the action distribution is piecewise constant, but the paper does not explain why this is the case (if I understand correctly, I guess it is because of the block-MDP structure?)
- Several propositions from the appendix, e.g. Proposition 9, Assumption 10, etc., are referenced without any indication of what they say or entail. (There is also a typo on Page 9 which says "Proposition 9 from the previous section"). Based on the numbering it is also not obvious they are located in the appendix. Any proposition or assumption from the appendix that is referenced in the main text should be referenced along with a statement saying it is in the appendix and a summary of what it says. (One suggestion would be to use numbering like Proposition A.1, where the "A" indicates appendix).
- I don't understand the point of Definitions 6 and 7 are, or what their relevance is for Theorem 8.
- The experiments section needs considerably more detail, enough to know at least the high-level experimental setups. I appreciate some of the details are in the appendix but there is both not enough even in the appendix and definitely not enough in the main text. Here are some specific things that were unclear to me (though this section should include more details in general, not just the specifics I point out here):
  - In Section 6.1, details need to be included regarding: the nine positions the arm can be in (i.e. describing Figure 3 in more detail), what policy is used to collect the 14,000 samples for training AC-State, and what the "counterexamples" mentioned in Figure 5 are.
  - In Section 6.2, what policy is used to generate the data for the top half of Figure 4? And as discussed above, the planning policy needs to be more clearly explained.
  - In Section 6.3, how are the various metrics (controllable state accuracy, exogenous noisy-ignoring accuracy) computed? How are the various baselines in Figure 6 implemented? e.g. what is the "contrastive" baseline---there are many contrastive methods, which one was used here?

Recommended changes:
- More intuition should be provided in the introduction about how/why AC-State works so that its relation to other methods can be better understood in the background section. I realize the introduction states *what* AC-State does---e.g. that it learns a multi-step inverse model with an information bottleneck---but it doesn't explain the motivation for this or *why* it works. The explanation right at the end of Page 7, for example, would be great to include earlier in the paper.
- I would recommend swapping the order of sections 2 and 3, reducing the length of sections 2 and 5, and lengthening section 6. The figures should also be reordered to come in order of mention in the paper: figure 1, 3, 2, 5, 4, 6.
- Definition 1, 5, 6, 7; Proposition 4; and Theorem 8 are never referenced again after being introduced. Definitions etc. should only be included if they are actually referenced. Please remove these or ensure they are properly referenced.
- The paper would be much stronger if experiments were included demonstrating how well AC-State performs when its assumptions are violated. Does it degrade gracefully or catastrophically? What are its failure modes?

**Strengths And Weaknesses:**

### Strengths

- The paper tackles an important problem, which is how to learn a low-dimensional representation from high-dimensional inputs while being invariant to exogenous noise.
- The method comes with theoretical guarantees, which is nice.
- The empirical evaluation is done across diverse environments: robotic control, multi-agent, and visually-rich navigation.

### Weaknesses

- I found the organization of the paper difficult to follow.
- The theoretical analysis is dense and appears to be included at the expense of more intuitive explanations for why/how the method works.
- There are many details missing about the experiments and how the method is implemented in practice.
- The difference between this paper and Efroni et al. (2022c) is unclear.
- The experiments do not provide insight into how the method compares to existing approaches that are also designed to ignore exogenous noise.
- The assumptions needed for the method to work are quite strong and it is unclear how the method behaves if these assumptions are not met.

---

> ### Author Response · Authors · 2022-12-27
> **Response Part 1**
>
> We appreciate that you liked the problem setting, the theory, and the choice of environments.  Thank you for your detailed and high-quality feedback.  While we believe that we have addressed your feedback to make the paper better, please don’t hesitate to suggest any additional experiments or changes you feel would make the paper better.
>
> Regarding your suggested changes, we think they’re quite good, and we’ve made the paper better by addressing them in our new revision:
>
> >I did not find Figure 1 to be particularly helpful in understanding the method. It's not obvious which parts are learned or how they are learned (e.g. what the losses are)
>
> We’ve replaced it with a new figure 1, which is more detailed and includes the losses.
>
> >Section 3 discusses a number of related approaches which attempt to discard exogenous noise, but dismisses them because they are not theoretically guaranteed to obtain the control-endogenous state
>
> Our claim is a bit stronger than this.  For many methods, there is a theoretical proof (or counterexample) showing that they won’t ignore exogenous noise.  For example, an autoencoder or a contrastive method should always capture exogenous noise.  This is more than just saying that there’s a lack of theoretical guarantee; rather, there’s strong theoretical evidence for failure under some assumptions.
>
> >Table 1 lists several methods with no explanation, description, or citations ... caption should also include citations
>
> Great point - we’ve fixed table 1 exactly as you suggested.
>
> >clarify the difference with Efroni et al. (2022c).
>
> The PPE paper (Efroni et. al. 2022c) uses open loop planning from a fixed start state (with strict episodic resets) with deterministic dynamics to do path prediction to merge different states.  The requirement of a fixed start state cannot be addressed merely through more computation (such as by training PPE from many different start states and merging the results).  This setting, which is required to run the algorithm at all (not just for theoretical guarantees of correctness), is much more restrictive than what AC-State requires.
> We suspect that AC-State would have better sample complexity because, once the agent has reached a goal state, it should be possible to pick a new state on the frontier that is close to the agent’s current position.  In other words, a state along the frontier of exploration is likely to be close to another state along the frontier.  This is not the main point of the method, though, so it would be more of a supplementary result.
>
> >information bottleneck and states ... method finds the lowest-capacity f. This claim is not explained or justified.
>
> We added more about this, along with an entire subsection on theory intuition (5.1) and a new figure giving some of the theory’s intuition (Figure 5).  The simplest argument we can make for the bottleneck is that it forces the representation to actually be learned in $f(x)$.  In the absence of any bottleneck, we could set $f(x)=x$ and still find the optimal multi-step inverse model solution $p(a_t | f(x_t), f(x_{t+k}); k)$.  A more technical explanation is that we need to establish that we find the coarsest (smallest) solution with f.  The other part of the theory (5.3) establishes that there is no coarser solution than the control-endogenous latent state.
>
> >L_bottleneck is not defined
>
> In our view, because this is a standard architectural element (the vq-bottleneck), we prefer to refer to the other paper and the github repo with the exact code.  We’ve updated the paper to make this clearer and explained the significance and intuition behind using the bottleneck. Additionally, if someone in the future finds a better way to learn discrete latent variables in a NN, it would be perfectly fine to use that in AC-State instead of the vq-bottleneck.  If you want us to also write out the loss terms in our paper as well, we can do that.
>
> >I was confused about how "AC-State with Planning Policy" works. Is the idea that you are given a planning policy, which is just used for data gathering to train AC-State? ... Or is it used in-the-loop with AC-State (in which case, are there any stability issues with training, that often occur in deep RL)?
>
> It’s the last thing that you mentioned; we use it in the loop on top of the representation actively being learned by AC-State. The planning policy is built in a learned tabular MDP using the latent states from AC-State (the tabular MDP is built using all counts seen so far).  We found it to be stable, but one interesting thing is that the latent state dynamics will look stochastic at the very beginning of training as it tends to overuse the codes some.  One thing is that we are always selecting goals to achieve high state coverage and are doing exact planning in a tabular model, so this resolves many of the sources of instability in DeepRL (i.e., no TD-learning, no bootstrapping, no random exploration, no dependence on sparse rewards).

---

> ### Author Response · Authors · 2022-12-27
> **Response Part 2**
>
> >Section 5.1 seems to be from a previous paper. Is it necessary to reproduce here? It also feels a little bit like math for math's sake---isn't it kind of by definition
>
> It’s reproduced here because it’s critical to understanding the paper.  I think that it’s not a trivial result, because, as far as we’re aware, it’s fairly unique to generalized inverse models.  For example, autoencoders, temporal-contrastive models, generative models, and forward models all fail to have this property.  This result also makes the story much clearer to me: we first show that the control-endogenous state is a solution to the multi-step inverse model, and then in the new theoretical result we show that there does not exist a coarser solution.
>
> >it seems that an important conclusion from this section is that the action distribution is piecewise constant, but the paper does not explain why this is the case (if I understand correctly, I guess it is because of the block-MDP structure?)
>
> By piecewise constant, we just mean that the multi-step inverse model requires no dependence on x beyond the true control-endogenous state.
>
> >Several propositions from the appendix, e.g. Proposition 9, Assumption 10, etc., are referenced without any indication of what they say or entail. (There is also a typo on Page 9 which says "Proposition 9 from the previous section"). Based on the numbering it is also not obvious they are located in the appendix.
>
> Thanks.  We’ve fixed it now so that the numbering reflects the section and is always consistent.
>
> >I don't understand the point of Definitions 6 and 7 are, or what their relevance is for Theorem 8.
>
> Hopefully the new Figure 5 and Section 5.1 on the theory’s intuition will help with this.  Basically, we have a notion of consistent partitions along with the set of all multi-step inverse models applicable over a certain horizon.  The basic idea of this part of the theory is we can produce certificates for all pairs of states which show that splitting distinct states can yield a reduction in the loss of the multi-step inverse model.
>
> >The experiments section needs considerably more detail
>
> We’ve added more detail about the experiments to the appendix, as you suggested.
>
> >I would recommend swapping the order of sections 2 and 3, reducing the length of sections 2 and 5, and lengthening section 6. The figures should also be reordered to come in order of mention in the paper: figure 1, 3, 2, 5, 4, 6.
>
> This is a good suggestion; done.
>
> >Definition 1, 5, 6, 7; Proposition 4; and Theorem 8 are never referenced again after being introduced. Definitions etc. should only be included if they are actually referenced. Please remove these or ensure they are properly referenced.
>
> I’m not sure why this would help, because we are still referencing the result, even if it’s not named.  For example, we introduce the reachable set R(s,h), and then just use this by name.
>
> >The paper would be much stronger if experiments were included demonstrating how well AC-State performs when its assumptions are violated. Does it degrade gracefully or catastrophically? What are its failure modes?
>
> Tables 3 and 4 in the appendix have been added to probe two potential failure cases: (1) insufficient maximum number of learned codes and (2) insufficient samples, and we show that both degrade the performance of AC-State fairly gracefully.

---

### Review · Reviewer_vkF8 · 2022-12-14

**Summary Of Contributions:**

The paper presents a simple yet effective approach to extract control-endogenous latent states (e.g., position of an agent) from complex observations and discard control-exogenous information (e.g., texture, lighting). The paper proposes a simple multi-step inverse model objective that predicts the first action given the current observation and a future observation. Information bottleneck is applied to the latent representation to achieve a minimal latent state. Theoretical analysis is performed to show that the minimal representation minimizing the multi-step inverse model objective is the control-endogenous state. Experiments are performed on robotic arms, maze exploration, and first-person indoor navigation, which demonstrate the proposed method’s superior performance over the baselines.

**Audience:**

Yes

**Claims And Evidence:**

Yes

**Requested Changes:**

- The claim “contains all of the information necessary for controlling the agent, while fully discarding all irrelevant information” is too strong. How do we know that the learned latent states “fully” discard all irrelevant information?
- Consistent baselines across all experiments.
- Discussion on how information bottleneck affects the theoretical analysis.

**Strengths And Weaknesses:**

**Strengths:**

- Learning an encoder that can obatin control-endogenous latent states from rich sensory input is important to achieve efficient RL in the real-world, where the true state of an agent is often unknown. The paper proposes a simple yet effective algorithm based on a multi-step inverse model objective with information bottleneck, which is easy to train with supervised learning and makes the approach scalable.
- The paper performs theoretical analysis to justify the multi-step inverse model objective.
- Empircally, experiments on three different tasks show that the proposed approach is effective and outperforms the baselines.
- The paper is well-written and the organization makes it easy to follow. The related work is thorough and clearly highlights the difference with prior work.

**Weaknesses:**

- The three tasks in the experiments use different baselines. To fully demonstrate the effectiveness of the approachs, the paper should compare with all baselines (especially stronger ones) used in the first-person navigation task on the other two tasks as well.
- It is unclear how the Information bottleneck used in the encoder affects the theoretical analysis in Sec. 5.2.
- The analysis assume determinstic dynamics, while in real-world scenarios, stochastic dynamics are common. The three tasks in the experiments also mainly have determinstic dynamics.

---

> ### Author Response · Authors · 2022-12-27
> **Response to Review**
>
> Thank you for your detailed feedback. We appreciate your insights and comments, which helped improve the paper significantly.
>
> We have added new experimental results to the paper, along with new baselines that are consistent across all experiments. We have also added new tables and figures to the appendix. We hope that the new findings highlight the importance of AC-State even more clearly.
>
> We also added more information about the information bottleneck and the theory.  The simplest explanation for why you need the bottleneck is that if you don’t have it, then $f(x)=x$ is a valid optimal encoder for $p(a_t | f(x_t), f(x_{t+k}); k)$, since there is no constraint placed on the multi-step inverse model itself.  The bottleneck forces the representation learning to happen in the encoder itself.
>
> >The analysis assume deterministic dynamics, while in real-world scenarios, stochastic dynamics are common. The three tasks in the experiments also mainly have deterministic dynamics.
>
> AC-State can be used in stochastic environments since it uses closed-loop planning and no part of the algorithm itself assumes deterministic dynamics.  However, most of the theory will not apply since the theory in 5.3 (that there is no coarser solution to the multi-step inverse model) assumes deterministic dynamics (it also easily generalizes to the special case of nearly-deterministic dynamics).  In the case of stochastic dynamics, one can construct counter-examples when using a random rollout policy where the multi-step inverse model cannot discriminate between stochasticity in the dynamics and stochasticity in the policy.  So generalizing AC-State to also work in stochastic environments would likely require us to be more careful about constructing the policy.  More specifically, we’d probably have to require that the policy always be a goal-seeking policy.  This approach was used in the OSSR paper (https://arxiv.org/pdf/2206.04282.pdf), which assumed access to high-level state variables but also was able to handle stochastic dynamics.
>
> Experimentally, we did try running the maze environments with stochastic sticky actions, such that on each step the agent stays in the same state (i.e., the action fails to execute) with a 20% probability.  AC-State still works and learns transition dynamics with roughly the right probability.  While this is simple, it would break PPE because open-loop planning does not work in this environment.  Even simple stochastic environments make the visualization of the learned state substantially more complicated (we can make such plots, but they’re hard for a reader to make sense of).
>
> >The claim 'contains all of the information necessary for controlling the agent, while fully discarding all irrelevant information' is too strong. How do we know that the learned latent states “fully” discard all irrelevant information?
>
> The claim is only under the realizability assumption (that we perfectly satisfy the objective).  The claim is correct asymptotically if the theory is correct (which we believe it is) and if the assumptions hold.  Especially because it’s an abstract, we want to emphasize the strength of the result in the theory, even though it does require many assumptions to hold.

---

### Review · Reviewer_SvYL · 2022-12-14

**Summary Of Contributions:**

 An agent interacting with an environment with dynamic background changes (noises) would need to separately model the controllable aspect of the state in order to correctly plan or learn robust policies. This work presents a method to learn agent-controllable latent states through learning a multi-step inverse dynamics model (i.e. predict actions given future observations).


**Audience:**

Yes

**Broader Impact Concerns:**

There is no ethical concerns to be addressed.

**Claims And Evidence:**

Yes

**Requested Changes:**

- Related work section: Compare AC-state discovery with the literature of causal learning / counterfactual learning. How would methods in learning causal dynamics model perform in Ex-BMDP? Is AC-state discovery a simpler problem than causal inference? If so, in what scenarios would AC-state be preferred over causal methods?

- Network architecture & Ablations: provide details of model design and loss terms used; show model performances ablating hyperparameters such as number of inverse dynamics steps and bottleneck state size;

- Limitations and future work: discuss the limitations of AC-state or plans for scaling AC-state to complex robot learning problems (i.e., demonstrate superior performance of learning with the latent space discovered by AC-state on downstream imitation or reinforcement learning tasks).


**Strengths And Weaknesses:**

Strengths:
- this work provides both intuitive explanations and a theoretical analysis of the method;
- the experiments provided in this work is neatly designed to showcase the problem setting where AC-state has an advantage over other methods; the maze environment is very interesting;

Weakness:
- lacking discussion of some related work in causal learning; the idea of learning agent-controllable states is very much related to the literature of learning causal models for robots (be it policies, dynamics models or state representation); prior work has explored this front a lot (to list a few representative ones see ref. [1-3]) and it is important to distinguish how AC-state is different from ideas in causal learning
- The empirical results are limited to learning reachable states and no downstream control task performance is shown; is the method limited to only these tasks because the assumptions of the AC-state are too strict? How would someone leverage AC-state for better downstream task learning?
- Is AC-state effectively learning to ignore noise in the environment? What if the noise is biased/spurious given the control? would AC-state still be able to recover the true invariant state?
- The implementation details are not sufficient for reproducing the results; what is your network architecture? What are the hyperparameters and how were they tuned? Was there any domain knowledge needed in tuning the model or do generic parameters work for a range of tasks?


[1] Mastakouri, A. A., Schölkopf, B., & Janzing, D. (2021, July). Necessary and sufficient conditions for causal feature selection in time series with latent common causes. In International Conference on Machine Learning (pp. 7502-7511). PMLR.

[2] Li, Y., Torralba, A., Anandkumar, A., Fox, D., & Garg, A. (2020). Causal discovery in physical systems from videos. Advances in Neural Information Processing Systems, 33, 9180-9192.

[3] Fu, X., Yang, G., Agrawal, P., & Jaakkola, T. (2021, July). Learning task informed abstractions. In International Conference on Machine Learning (pp. 3480-3491). PMLR.

---

> ### Author Response · Authors · 2022-12-27
> **Response Part 1**
>
> We appreciate that you liked our theory, intuitions, and proposed environments.
>
> We completely agree about causal learning.  We’ve added a new subsection to the related work just talking about causal learning, and we’ve added citations for all the papers that you mentioned in your review (although one of the papers we put under the reward-based learning section).
>
> The connection between AC-State and causality is very interesting.  We think the problem of learning a causal model from observations and actions can be divided into two parts: representation learning and learning the causal dynamics.  One of the distinct aspects of AC-State is that it learns a representation of the control-endogenous state from rich observations (such as images).  There is no simple variable-wise selection that extracts the control-endogenous state.  An example of this is the robot arm, where some pixels can be the robot arm on some time steps or the television screen on other time steps.  Moreover, the representation is learned from scratch without any use of a pre-trained representation.  This differentiates AC-State from most causal learning work, in which some high-level representation is already given.  For example, Causal Dynamics Learning (Wang 2022) assumes access to high-level variables and performs conditional independence testing over these variables to discover the causal dynamics.
>
> One way in which the causal learning literature goes further than AC-State is that many papers learn a significantly richer factorized structure than the Exogenous Block-MDP structure.  For example, it would be interesting to distinguish between factors that are directly controllable by the agent and factors that are relevant for the agent’s control.  For example, the weather is not controllable but is control-relevant.  A richer causal structure could enable better planning and exploration.  The Causal Dynamics Learning Paper (Wang 2022) distinguishes between these two kinds of states.  In our view, combining the representation learning capability of AC-State with a richer kind of state factorization is a very promising area for future research.
>
> Regarding downstream tasks, our view is that AC-State is trying to tackle a more general problem of discovering the control-endogenous state and its associated dynamics.  If we can do this correctly, then we can solve any “task” in the environment by using value iteration to plan to reach a goal state without having to collect any additional data from the environment.  At the same time, the idea of solving a more natural and abstractly defined set of tasks in an environment will make more sense when we move past the tabular-MDP setting, which is something that we’re excited about for future work.
>
> To scale to more complex robotic environments, we would need to move away from tabular-MDP to handle compositional and continuous states.  If we simply make the hidden state continuous, we can use something like AC-State and still get most of the relevant information, but in an unstructured form.  The issue that arises is that when we want to do planning and exploration, we no longer have the provably good algorithms that are available in the tabular setting.  Moreover, we can’t guarantee exact state coverage, so the theoretical assumptions of AC-State would be severely violated.
>
> We think we could do something where we use a goal-conditioned version of MuZero to plan along with BYOL-Explore to define rewards and a variant of AC-State to learn the representation while doing exploration, but this would make the experiments only loosely inspired by the theory in the AC-State paper.  We’re actively pursuing and very interested in these directions nonetheless.

---

> ### Author Response · Authors · 2022-12-27
> **Response Part 2**
>
> “Is AC-state effectively learning to ignore noise in the environment? What if the noise is biased/spurious given the control? Would AC-state still be able to recover the true invariant state?”
>
> The best way to think about it is to look at the factorization assumption (Proposition 3).  Our latent dynamics factorize into p(s’ | s,a) and p(e’|e), and we are able to learn to ignore p(e’|e) as well as the emission distribution q(x | s,e).  So effectively, we can ignore any noise that can be factored away from the actions. This includes things like background distractions, visual details, and graphics.
>
> Your second point is pretty interesting.  If we understand correctly, you’re thinking that there may be a noise signal where the characteristics of the noise are dependent on the action, for example, a noisy visual animation where the choice of animation is influenced by the agent’s actions.  We think that what happens here is that the choice of noise as an index variable is in the control-endogenous state, but the noise itself remains exogenous.  Our more general advice for trying to find such a factorization conceptually is to write out what you think the causal graph is with high-level variables, and then try to group them to put as many into an exogenous state “e” such that p(e’ | e) factorizes.  We talk about this some in Appendix F, and it’s also nicely discussed in the Wang 2022 et. al “Causal Dynamics Learning” paper (https://arxiv.org/abs/2206.13452)

---

### Comment · Action_Editors · 2022-12-19
**Likely prolonging discussion due to holidays**

I'd encourage the authors to reply to the reviewers as promptly as possible since the official timeline would have the 2 week discussion period last until Dec 28th, at which point reviewers are supposed to finalize their decisions.

However, I'll note that I don't think it is reasonable to expect the reviewers to be obligated to respond promptly during the holiday period.  So rather than the standard 2 weeks, I'm going to not expect reviewers to finalize their decisions until January 6th (an extra week and two days).  If anyone objects to this, let me know.  And if reviewers do submit final decisions before that, we can potentially proceed more quickly.

---

### Author Response · Authors · 2022-12-27
**Summary of Revisions to Draft and Major Improvements**

We thank the reviewers for their excellent feedback.  Many reviewers praised the paper for the theoretical justification (SvYL, vkF8, jojo), the effectiveness of the proposed approach in experiments (SvYL, jojo) and the importance of the problem of learning the control-endogenous latent state (vkF8, jojo).

At the same time, reviews brought up some important issues, which we’ve addressed by revising the paper and adding substantial new content.

	(1) We added a completely new main figure, which explains the method more clearly. Figure 1 is now more scientific and detailed.  It clearly shows the objective and the exogenous Block-MDP setting. We hope this new figure provides better insights to the reviewers about why the AC-State objective is useful in the specific exogenous setting that we consider, assuming factorized dynamics.

	(2) The organization was revised some with the related work section introduced earlier in the paper, and a new part specifically on causal learning has been added to related work. We also provided additional details and comparisons to previous works (e.g Efroni et al., 2022). The related work table-1 has also had citations added to make it easier to read. We hope the additional details and updates in the related work section will provide more insights to the reviewers about the significance and contribution of the AC-State objective and why it is useful for recovering the control-endogenous state compared to prior works.

	(3) Additional Experimental Results: We included more baselines for the maze and robot arm tasks, as well as more ablation experiments to investigate the effect of changing hyperparameters.  Additional results are provided in tables 2,3,4, and 5 in the appendix, along with a new Figure 8 showing more visualizations of baseline performance for the robot arm.

	(4) Reviewers rightfully pointed out that we could have made the theory more intuitive and explained the role of the bottleneck better.  We added a new subsection 5.1, adding intuition for the theory, along with a new illustration of how the proof works in Figure 5.

---

### Decision · Action_Editors · 2023-01-16

**Recommendation:** Accept with minor revision

**Comment:**

All reviewers agreed that the motivation for the AC-state innovation presented in this work was clear and the combination of theoretical analysis and empirical investigation is compelling.  However, the reviewers were split as to whether the work is sufficiently clear to accept in its current form.  Reviewer jojo found the motivation and core contributions of the work compelling (the problem, the theory, and the experiments).  However, reviewer jojo also found the work poorly organized and a bit unclear throughout.  The reviewer identified a number of specific clarity issues.  The authors made an attempt to address some of the issues raised by the reviewer, which the reviewer acknowledged, but the final response of the reviewer is that the changes were not comprehensive enough.  The reviewer included some precise final points and emphasized their criterion for clarity being: "Is there enough detail in the main text of the paper that someone could implement the overall structure of the method and experiments...and is there enough detail in the appendix that they could get all the details too".  On the other hand, reviewer vkF8 found the paper "well-written and the organization makes it easy to follow".  The reviewer requested that the authors include consistent baseline algorithm comparisons for each of the 3 experimental settings, which the authors did.  However, consistent with the spirit of reviewer jojo's concerns, the presentation of the new experimental results does not have a consistent format, with very little results in the main text.  Reviewer SvYL also noted that implementation details in the initial submission are not sufficient for reproducing the results, and it was not clear if the authors addressed this specific point.

Summarizing the reviewer feedback, informed by reviewer jojo's points, I think the current draft needs something that could be either framed as minor revisions or a bit more than that, depending on the perspective.  Since the residual issues center on clarity and presentation, I do not think it is a good use of reviewer time for this paper to be re-reviewed separately to address this, so I propose to accept this with requested revisions, that I will check for before approving the final version.  I would encourage the authors to make as thorough a pass at revisions as they can, informed by reviewer jojo's additional feedback.  I'll produce a digest of a few specific points below that should be treated as a minimal set of required changes.

Some specific areas to focus revisions around:
- The point being made by figure 2 could be clearer. Edit the figure and/or caption.
- Section 4 introduces AC-state quite generically, but parameter choices made for each experiment type are lacking (as noted by reviewer SvYL).  As a concrete example, for each of the three experiments, the horizon parameter should be discussed.
- The empirical results figures (currently figures 4, 6, and 7), are awkwardly placed in the paper (as noted by reviewer jojo), with figure 4 coming way too early and the order of maze and robot figures switched(?).  These figures seem like they should occur within section 6.
- More clarity around the algorithm in the main text would be useful (include a version of the algorithm box, or equivalent information, in the main text).
- Each of the three experiment types should have a consistent summary table with core metrics in the main text. As previously requested by the reviewers, the baselines are the same, but there is inconsistency in how and where the results are presented.
- Overall, aim to make the main text reasonably self-contained, with additional detail in the supplement.  As reviewer jojo pointed out, some rather essential information is presently in the supplement.
- Address jojo's remaining major points: algorithmic clarity around when updates happen relative to agent sampling, enumerate the baseline choices clearly in the main text, experimental details including number of seeds in the main text, etc.

**Audience:**

The material is suitable for the TMLR audience.

**Claims And Evidence:**

The claims and evidence appear to be accurate and convincing.  However, there is reviewer disagreement about whether the presentation is sufficiently clear for acceptance.